# Innovating within or outside dominant food systems? Different challenges for contrasting crop diversification strategies in Europe

**Kevin Morel**[1,2]*, **Eva Revoyron**[2,3], **Magali San Cristobal**[4], **Philippe V. Baret**[1]

**1** SyTra, Earth and Life Institute, UCLouvain, Louvain-la-Neuve, Belgium, **2** UMR SADAPT, INRAE, AgroParisTech, Université Paris-Saclay, Paris, France, **3** USC LEVA, INRAE, Ecole Supérieure d'Agricultures, Angers, France, **4** UMR 1201 DYNAFOR, INRAE, Université de Toulouse, Toulouse, France

\* kevin.morel@inrae.fr

**Data Availability Statement:** Data files are available from the Zenodo database (https://zenodo.org/record/3249967#.XQoQoo_gpPY)

**Funding:** This project has received funding from the European Union's Horizon 2020 research and

## Abstract

Innovations supporting a shift towards more sustainable food systems can be developed within the dominant food system regime or in alternative niches. No study has compared the challenges faced in each context. This paper, based on an analysis of 25 cases of European innovations that support crop diversification, explores the extent to which barriers to crop diversification can be related to the proximity of innovation settings with dominant food systems. Drawing on a qualitative analysis of interviews and participatory brainstorming, we highlight 46 different barriers to crop diversification across the cases, at different levels: production; downstream operations from farm to retailing, marketing and consumers; and contracts and coordination between actors. To characterise the diversity of innovation strategies at food system level, we introduce the concept of "food system innovation settings" combining: (i) the type of innovative practice promoted at farm level; (ii) the type of value chain supporting that innovation; and (iii) the type of agriculture involved (organic or conventional). Through a multiple correspondence analysis, we show different patterns of barriers to crop diversification according to three ideal-types of food system innovation settings: (i) "Changing from within", where longer rotations are fostered on conventional farms involved in commodity supply chains; (ii) "Building outside", where crop diversification integrates intercropping on organic farms involved in local supply chains; and (iii) "Playing horizontal", where actors promote alternative crop diversification strategies—either strictly speaking horizontal at spatial level (e.g. strip cropping) or socially horizontal (arrangement between farmers)–without directly challenging the vertical organisation of dominant value chains. We recommend designing targeted research and policy actions according to the food systems they seek to develop. We then discuss further development of our approach to analyse barriers faced in intermediate and hybrid food system configurations.

## 1 Introduction

Transition toward more sustainability requires changes in food systems [1–4]. While agroecology scholars initially emphasised the need for sustainable innovations at the farm level [5–7], social movements and researchers have increasingly acknowledged the necessity to analyse

                    

innovation programme (https://ec.europa.eu/programmes/horizon2020/en) under grant agreement No 72748. The funders had no role in study design, data collection and analysis, decision to publish, or preparation of the manuscript.

**Competing interests:** The authors have declared that no competing interests exist.

and support innovations at all levels of the food system [8–10], including all the actors, infrastructures, processes and activities related to the production, transport, processing, distribution, and consumption of food [11].

The adoption of more sustainable farming practices at a large scale is limited because the dominant food systems are shaped by a locked-in socio-technical regime [12–16]. The concepts of socio-technical regime and lock-in originated respectively from the multi-level perspective theory [17–19] and from economic studies on technological change [20, 21]. A dominant food system regime in a given biophysical, infrastructural and institutional context "represents mainstream social and technical elements dominated by conventional industrial farming and value chains controlled by large-scale and powerful agri-food industries and companies" [2]. The food system regime relies on a coherent set of procedures, cognitive routines, existing technologies, rules, skills, and values which shape the action of the food system actors from production to consumption [19, 22]. All those elements have co-evolved historically (path-dependency) and reinforce one another, resulting in the system's perpetuation and stability (lock-in).

Transition studies traditionally draw a contrast between incremental innovations, which are partial adaptations carried out by actors within the dominant regime, and radical innovations, which are developed and tested in niches created by outsider networks, guided by strong alternative values and/or new performance criteria [18, 19, 22, 23]. Many agroecology-related scholars have argued that sustainability issues require radical innovations, and have focused on ways to support alternative niche food systems such as organic farming, and to create appropriate conditions to transform dominant food regimes [1, 9, 22, 24–28]. However, radical innovation niches that support more sustainable food systems can also be developed by actors within the dominant food system [15, 29]. For example, the industrial French processor Valorex organised the coordination of mainstream value chain actors to promote omega-3-rich products from animals fed with linseeds (an innovative crop improving agricultural rotations). This was done through the dedicated Bleu-Blanc-Coeur label. Another case in point is the Qualisol cooperative which developed large equipment and invested heavily in R&D to encourage lentil-wheat intercropping with guaranteed minimal prices to secure the incomes of farmers taking the related risks [30].

To what extent are the barriers to sustainable innovation at different levels of food systems dependant on innovation settings? To our knowledge, no study has compared the challenges of innovating within or outside the dominant food systems, with similar objectives. Most sustainable agriculture transition studies have explored the challenges for innovations designed either outside the dominant food systems [1, 22, 24, 26–28] or within them [14–16, 30, 31].

In this paper we investigate the above question, analysing 25 European innovation cases in which crop diversification was promoted. Crop diversification is a pillar of agroecological transition [16, 32] that has been less studied than other sustainable agriculture approaches such as pesticide reduction or organic farming, and never in a European context.

Our analytical framework is inspired by (i) concepts from the multi-level perspective on socio-technical transitions [13–19] to investigate innovation settings in relation to their proximity with the dominant regime; and (ii) recent agroecology literature arguing that supporting innovation in food systems requires a preliminary characterisation of their structure and different interrelated components [2, 33]. We introduce the concept of "food system innovation settings" (see Section 2.1) to characterise the diversity of innovation strategies at food system level. Through a multiple correspondence analysis (MCA) we are able to link specific combinations of barriers to crop diversification, on the one hand, to the proximity of innovation settings with the dominant food system regime in terms of agricultural practices and type of value chains, on the other. Based on the three ideal-types of food system innovation settings,

we suggest that targeted research and policy actions be designed according to the food systems they seek to develop. We discuss further development of our approach to analyse barriers in intermediate and hybrid food system configurations.

## 2 Material and methods

### 2.1 Contrasting food system innovation settings for crop diversification in Europe

Crop diversification faces a typical situation of socio-technical lock-in of dominant food regimes [16]. It could be a pillar of agroecological transition, as it potentially has multiple sustainability benefits, such as reduced yield gaps and dependency on external inputs, increased biodiversity, limited economic risks, more varied landscapes, and enhanced provision of diverse ecosystem services [16, 32, 34]. Despite such well-known benefits, various studies have shown the adoption of diversification practices to be limited by barriers at all levels of food systems. These barriers are related to "regime rules" [19] historically established to support large-scale specialisation and short-term maximisation of profits with chemical inputs [16, 31, 35–38]. We posit that crop diversification is specifically relevant as it can be promoted both outside and within the regime [2, 30].

Our research is based on 25 cases of innovations promoting crop diversification at the food system level in 11 European countries (Table 1) within the DiverIMPACTS multi-actor project (https://www.diverimpacts.net/). In line with the rationale of this European project, the 25 cases were initially selected to cover a wide range of situations as far as farms' pedoclimatic conditions and diversification strategies were concerned. This initial selection did not take into account the type of value chain and/or agriculture (organic or conventional), which explains that the cases design is not optimal with regard to the variables considered in this study.

Each case was led and monitored by an "innovation team" of two local actors who were in charge of stimulating collaboration between research bodies, farmers, farmers' organisations, associations, industries, businesses and public institutions.

The "new" crops promoted in each case were widely diverse in terms of context and objectives, ranging from leguminous crops such as alfalfa, clover, soybean, lentils, lupine, various types of peas and beans, to oleic crops such as rapeseed, hemp, sunflower, gold-of-pleasure (*Camelina sativa*) and milk thistle (*Silybum marianum*), to minor grain crops (at least in Europe) such as sorghum, buckwheat, quinoa, and millet, and finally vegetables. Each case focused on one or more diversification crop, either to be used for animal feed, or for human food, or both.

To account for the diversity of innovation strategies and contexts, we define "food system innovation setting" as the combination of: (i) the type of innovative practice promoted at the farm level; (ii) the type of value chain supporting that innovation; and (iii) the type of agriculture involved (organic or conventional). This definition echoes the characterisation of food systems as interrelated components [2, 39] of: (i) the agricultural production system; (ii) the value chain; and (iii) the support structures (advisory, R&D, innovation policy, etc.). In our case, the type of agriculture is related both to the agricultural production system and to the support structures which often differ for organic and conventional agriculture.

Among the 25 cases, we characterised three categories of diversification practices, value chains and types of agriculture, based on a preliminary qualitative analysis of interviews with innovation teams (see part 2.2). These categories were tailored to reflect the preliminary links that we found between the diversity of innovation settings and existing barriers (before running the multiple correspondence analysis described in Part 2.3).

The three different innovative farming strategies considered were (Fig 1):

**Table 1. Characteristics of the 25 food system innovation settings of crop diversification in Europe.**

| Diversification farming strategy | Value chain | Agriculture type | Country | Case ID |
|---|---|---|---|---|
| Temporal | Commod | Including conv | Germany | 3 |
| Temporal | Commod | Including conv | Romania | 8 |
| Temporal | Commod | Including conv | France | 13 |
| Temporal | Commod* | Including conv | France | 25 |
| Temporal | Local | Only organic | Switzerland | 6 |
| Temporal | Local | Only organic | Hungary | 7 |
| Temporal | Local | Only organic | Poland | 10 |
| Temporal | Local | Only organic | Netherlands | 23 |
| Temporal | Arrang | Including conv | France | 11 |
| Temporal | Arrang | Only organic | Belgium | 21 |
| Spatial | Local | Including conv | Italy | 9 |
| Spatial | Local | Including conv | Italy | 22 |
| Spatial | Local | Only organic | Netherlands | 16 |
| Spatial | Arrang | Including conv | United Kingdom | 2 |
| Spatial | Arrang | Including conv | Belgium | 4 |
| With intercrop | Commod | Including conv | France | 5 |
| With intercrop | Commod | Including conv | Belgium | 12 |
| With intercrop | Commod | Including conv | France | 14 |
| With intercrop | Commod | Including conv | Belgium | 17 |
| With intercrop | Local | Including conv | United Kingdom | 15 |
| With intercrop | Local | Only organic | Belgium | 18 |
| With intercrop | Local | Only organic | Sweden | 19 |
| With intercrop | Local | Only organic | Switzerland | 20 |
| With intercrop | Local | Only organic | United Kingdom | 24 |
| With intercrop | Arrang | Including conv | Netherlands | 1 |

Diversification farming strategy: [Temporal]: only temporal crop diversification; [Spatial]: including spatial crop diversification with no separation of harvested crops; [With intercrop]: including intercropping.

Value chain: [Commod]: crop diversification products target the commodity market and/or the agroindustry; [Local]: diversification products target local markets at the regional or national level; [Arrang]: diversification crops produced by plant producers are used by livestock farmers at the local level (direct arrangement between farmers).

Agriculture type: [Including conv]: the case includes conventional farmers (10 cases involve both organic and conventional farmers); [Only organic]: only organic farmers are involved in the case (no conventional farmers).

Modalities of variables describing the innovation setting were designed to ensure at least 5 occurrences of each.

*In that case commodity and local value chains coexist. However, during interviews, the innovation teams focused on the barriers linked to long value chains.

The cases are called by their ID in the DiverIMPACTS project, to make it easier to find more information on the project website: https://www.diverimpacts.net/case-studies.html

The description of the different cases corresponds to their innovation strategy in the initial phase of the project (late 2018, early 2019) when interviews were carried out. Some of them have since evolved.

- **Temporal crop diversification**: new crops are integrated into the crop rotation before or after another crop, either as a main cash crop (e.g. introducing sunflower after maize), as a fodder crop (e.g. introducing a 2-year cycle of alfalfa before wheat) or as a winter cover crop (e.g. sowing a plant mixture including leguminous crops to cover the soil and absorb nutrients during winter after a crop harvested in summer and before sowing a crop in the spring);

- **Spatial crop diversification** (with no separation of harvested crops): new and/or pre-existing crops are combined at the same time on the same plot to increase biodiversity and spatial

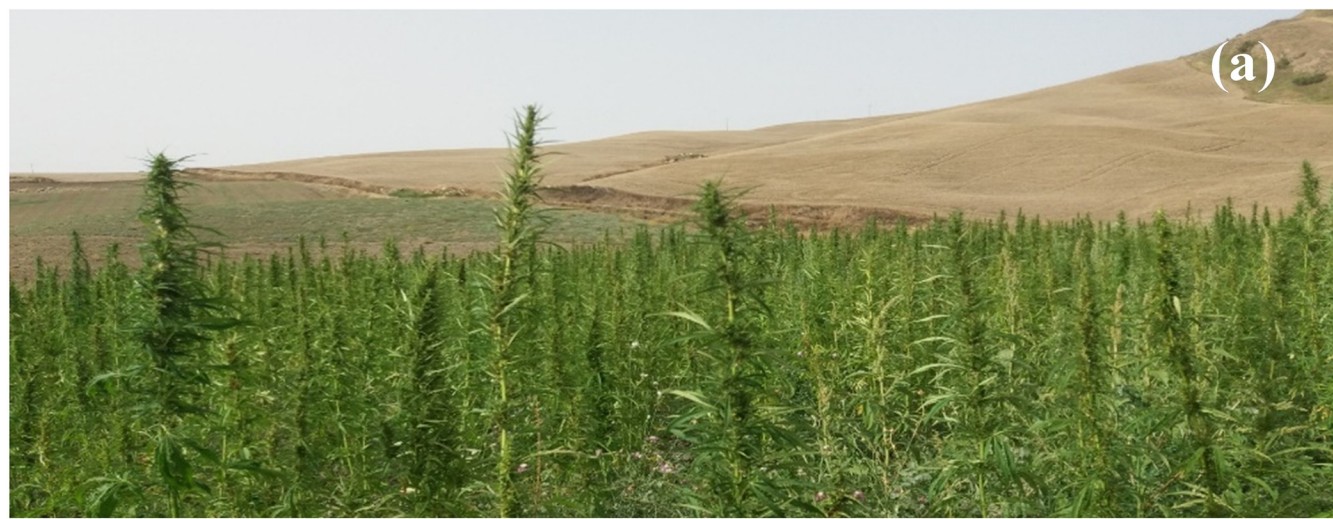

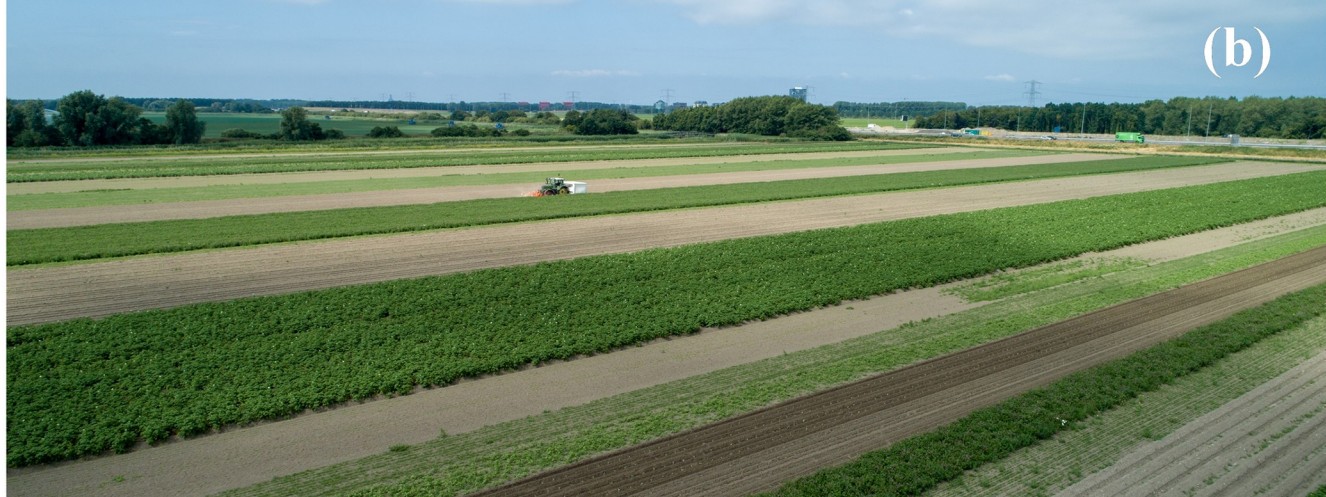

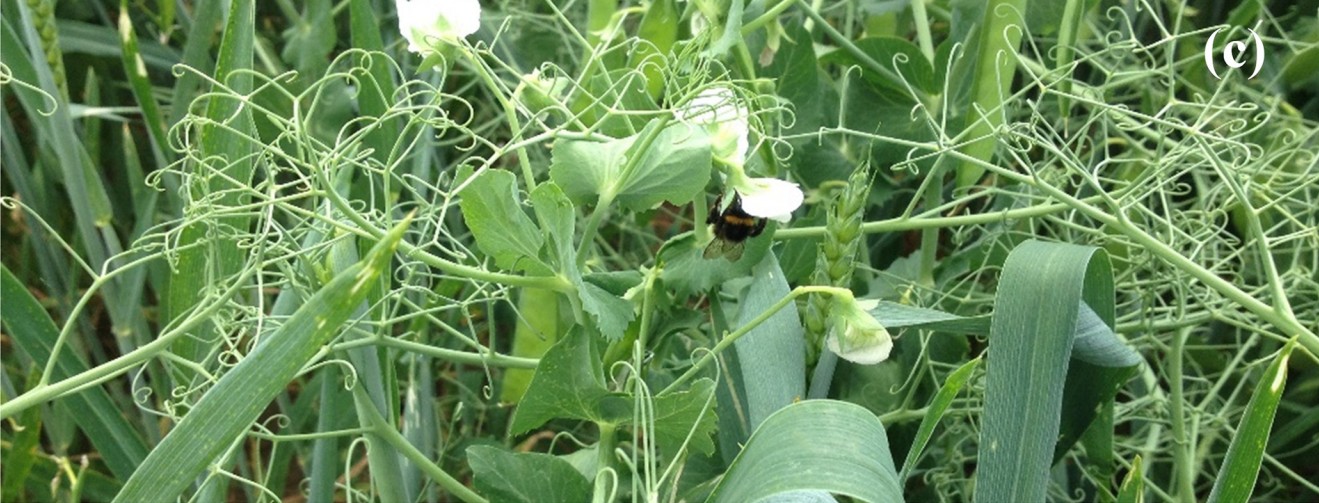

**Fig 1. Temporal diversification integrating hemp (longer rotations) in Sicily (a), strip cropping (spatial diversification) in the Netherlands (b), intercropping of wheat and winter pea in Belgium (c).** Credit: Luca Colombo (a), Bionext (b), Walagri (c).

heterogeneity at the field level. The cases in our study involved either plant mixes (e.g. new types of leguminous crops were mixed and sown together with winter cover crops) or strip cropping [40], in which crops are combined in distinct strips (e.g. 6, 12, 24 or 48m wide strips of grass/clover, carrots, potatoes, onions, Brussels sprouts in a field where normally only one crop will be grown). Crops do not need to be separated because they are either used together after harvest (e.g. as a fodder mix), are not harvested (e.g. in the case of a multi-species winter cover crop that is ploughed into the soil to build up fertility before sowing a new crop), or are harvested separately (in the case of strip cropping).

- **Intercropping**: we use intercropping here in a narrow sense to refer to a specific case of spatial crop diversification where crops are grown and harvested together, and then need to be separated to be used separately (e.g. winter pea and wheat sown at the same time in the same field).

Cases with spatial crop diversification or with intercrops often implement *de facto* temporal diversification because the new intercrop spatial configuration is integrated into a longer rotation. We chose to distinguish intercropping from other types of crop spatial diversification based on the need to separate crops after harvest. This was because both previous literature [35, 41] and qualitative analysis of our interviews show that crop separation introduces specific barriers into the cropping design, management and post-harvest phases.

To sell the products of crop diversification, the actors relied or sought to develop value chains oriented towards three types of outlet:

- **Commodity market**: products target long chains and possible export, often involving big agro-industry players. Localness is not an objective as such;

- **Local market**: products target local markets at the regional or national level with shorter chains involving essentially small-scale processing, and promoting alternative food systems;

- **Arrangement with livestock farmers**: products are used as feed (after harvesting or directly through on-plot grazing) by local livestock farmers through direct transactions based on a crop-livestock integration logic [42, 43]. Land exchanges can also be involved, e.g. when a livestock farmer uses arable farmland for 1 or 2 years to grow a grassland (bringing plant diversity into the arable crop rotation), while a livestock farmer grows cereal crops on arable land (for plant diversity between two grassland phases).

We also characterised cases according to the agriculture type: involving only conventional farmers, only organic farmers, or both (Table 1). This categorisation is based on the assumption that conventional farmers may face more barriers to crop diversification at the farm level because the development of conventional agriculture since World War II has relied on specialisation. Conversely, crop diversification has always been a pillar of organic agriculture.

## 2.2 Characterising barriers to crop diversification

To characterise barriers to crop diversification in each of the 25 innovation settings, we followed a multi-step procedure described in Fig 2.

In a first round, 5 workshops were organised, each involving 5 different innovation teams out of 25. Each team was involved individually in a 2-hours brainstorming (Fig 3a) based on the drawing of "problem trees" [44, 45]. This method aimed at investigating the different barriers that could limit or impede the diversification process, and the causes behind the actual

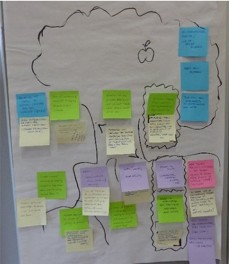

**25 cases of food system innovation settings in Europe**

This work relied on crop diversification dynamics promoted in 25 contrasted food system innovation settings (Table 1). Each case was led and monitored by an innovation team of 2 people.

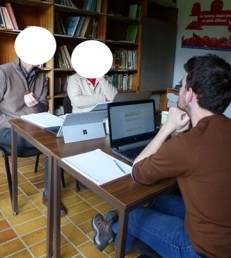

**Drawing of a problem tree by each innovation team**

A 2-hours brainstorming was carried out using the causal analysis method to draw a "problem tree" [44, 45]. This method aimed at investigating the different barriers that could limit or imped the crop diversification, looking systematically for the causes behind the current difficulties faced in each case.

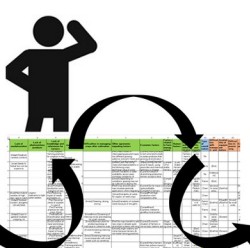

**Complementary interview with each innovation team**

The description of barriers was deepened in each case ensuring that all levels of value chain were covered: (i) farming and production (also integrating upstream aspects such as seeds or inputs availability), (ii) downstream operations from farm to retailing, (iii) marketing and consumers, (iv) contracts and coordination between actors [16].

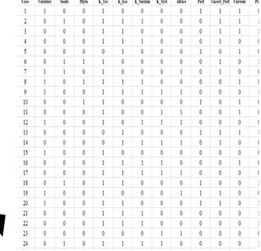

**Qualitative analysis of barriers across and within cases**

All barriers mentioned by innovation teams were grouped in more and more generic categories based on an iterative cross-analysis of interview content using thematic coding and matrix tools [46, 47] in the specific context of a multiple-case study [48, 49].

**Matrix of presence/absence of barriers to crop diversification in each case**

At the end of the process, 46 categories of barriers to crop diversification were stabilised (Table 2). The matrix with the presence/absence of barriers across the 25 innovation settings was the basis of the following MCA (Figure 3).

**Fig 2. Presentation of the different methodological steps involved in the characterisation of barriers to crop diversification across the 25 European food system innovation settings.** The individual appearing here has given written informed consent (as outlined in PLOS consent form) to publish these case details.

difficulties faced in each context. During the brainstorming exercise we took notes on the discussions within each innovation team and on the information drawn on the "problem tree". A synthetic report was also drawn up by each team to provide an overview of the issues identified.

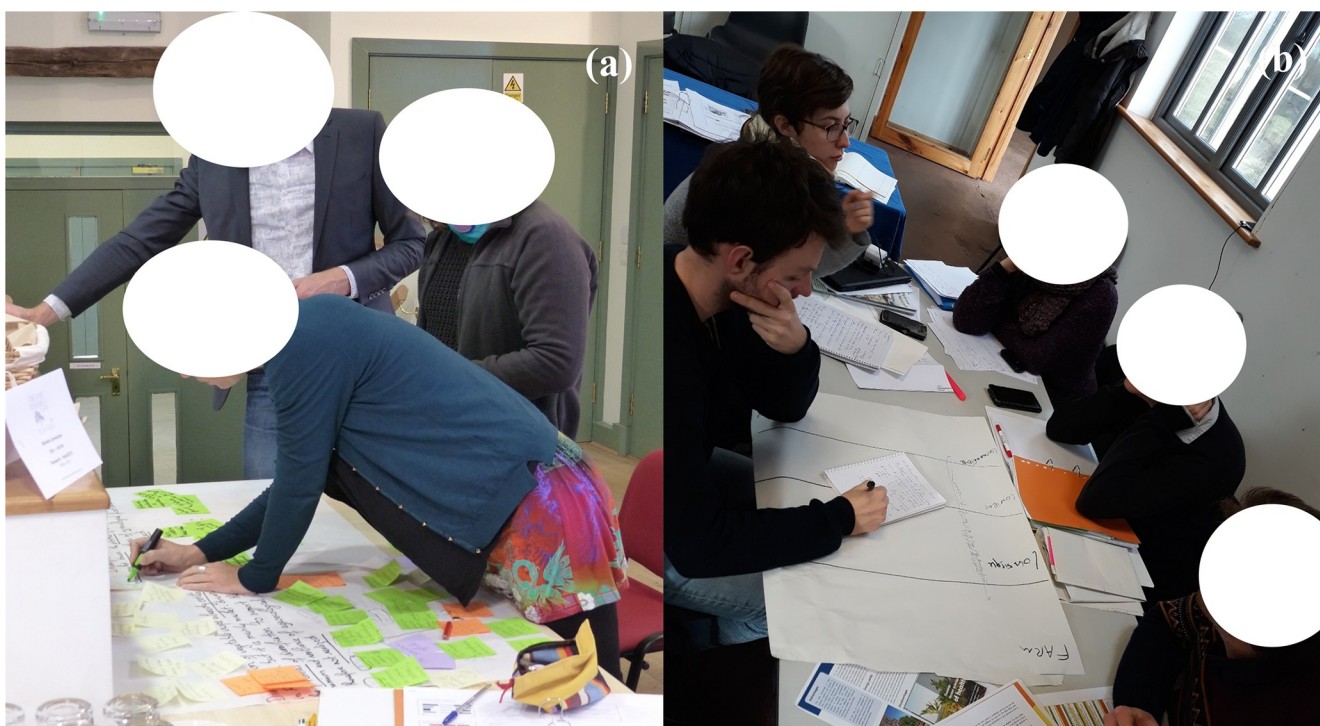

**Fig 3. An innovation team brainstorming on a "problem tree" in the first-round workshop (a) and a complementary interview in the second-round workshop(b).** Each innovation team consists in 2 participants (case leader and monitor). Additional persons are scientists facilitating the process or carrying out the interview. The individuals appearing here gave us written informed consent (as outlined in PLOS consent form) to publish these case details.

In a second round of workshops (4 months later in average), a short 30-minute complementary interview (Fig 3b) was carried out with every innovation team. Note that innovation team leaders and monitors mostly belonged to farming-related organisations (farmer's associations, agricultural R&D, extension or advisory services), which may create a bias in the perception of barriers at different levels of the value chain (see Discussion). To mitigate that bias, innovation teams were encouraged, before complementary interviews, to contact as many value chain actors as possible (farmers, processors, retailers etc.) in order to collect information about barriers at different levels of the food system. To support innovation teams in that task, we provided them with guidelines and a structured data collection framework (S1 Appendix).

During second-round complementary interviews with innovation teams, participants were asked to deepen the description of barriers to diversification at different levels of value chains. Interviewers ensured that all levels were covered: (i) farmers and production; (ii) downstream operations from farm to retailing; (iii) marketing and consumers; and (iv) contracts and coordination between actors. Those 4 categories, with a specific focus on coordination, were inspired by a previous study on factors impeding crop diversification [16]. We aggregated the notes taken during first-round brainstorming and second-round complementary interviews, as well as the first-round reports, to build the final "qualitative material" of our study. We then carried out qualitative analysis of this material using thematic coding and matrix tools [46, 47]. The general aim of this approach was to build more and more abstract categories on the basis of an iterative cross-analysis of interview contents from the multiple cases [48, 49]. This resulted in the categorisation of 46 barriers (Table 2) to crop diversification. The level of abstraction to characterise barriers was linked to the level of precision used by innovation teams to describe them. For example, innovation teams very specifically discussed many aspects of farmers' lack of knowledge and references. This is why we chose to distinguish 3 specific categories of barriers related to this lack of knowledge: technical implementation of farming practices (K_Tec); impact of new practices on the sustainability of the farm (K_Sustain); and impact of new practices on the global design of the farming system (K_Syst). Conversely, for other barriers, innovation teams mentioned very generic challenges with limited precision, which resulted in broader categories. For example, we considered a single category for challenges related to the Common Agricultural Policy (CAP) environmental or sanitary regulations, even though it may embrace various fields of application and impacts.

We built a matrix with the presence/absence of 46 barriers (Table 2) across the 25 innovation settings. The final dataset presenting barriers to crop diversification according to the characteristics of the 25 cases is available at: https://zenodo.org/record/3249967#.XQoQoo_gpPY

This dataset was the basis of the following MCA.

All interviewed human participants belonged to organisations which were partners of the DiverIMPACTS project. This study was approved by the Executive Committee of the DiverIMPACTS project (European Union's Horizon 2020 research and innovation programme under grant agreement No 72748). Consent was obtained orally and data analysed anonymously. The content of this paper and the related use of case-study data was submitted to all DiverIMPACTS partners allowing complaints or modification requests before submission.

## 2.3 Analysing links between barriers and innovation settings

Our objective was to explore the extent to which barriers to crop diversification could be specifically related to contrasting food system innovation settings. We performed a multiple correspondence analysis (MCA) of the presence/absence of the 46 barriers to crop diversification that we had highlighted across the 25 cases (with the following R-packages: FactoMineR [50,

**Table 2. Occurrence of barriers to diversification in the 25 case studies and their links to the ideal-types of food system innovation setting.**

| | Barriers to crop diversification | Code | n | W | O | H |
|---|---|---|---|---|---|---|
| **Agricultural production** | Lack of technical knowledge and references | K_Tec | 21 | | | |
| | Lack of economic knowledge and references | K_Eco | 16 | | 0 | 1 |
| | Need for investment for adapted machinery | Machin_Invest | 13 | 0 | 1 | 0 |
| | Lack of knowledge and references about impacts on sustainability | K_Sustain | 12 | 1 | 0 | |
| | Profitability is low, problematic or uncertain | Profit | 11 | 0 | | 1 |
| | Uncertainties, risks and variability of agronomic performances | Uncert_Perf | 10 | | | |
| | Lack of technical knowledge on the impact on farming systems and design | K_Syst | 9 | 1 | 1 | |
| | Lack of information because of problems with advisory context | Advice | 9 | 1 | | |
| | Current situation is still profitable in the short term | Current | 9 | 1 | | |
| | Constraints in labour organisation (period, volume), mental or physical load | Work | 9 | | 0 | 1 |
| | Barriers related to CAP*, environmental or sanitary regulations | Reg | 9 | | | 1 |
| | Lack of adapted plant varieties in the local context | Varieties | 8 | | | |
| | Need for innovation in machinery for field activities | Machin_Innov | 8 | 1 | 0 | |
| | Low agronomic performance (yield, quality) | Perf | 8 | | 1 | |
| | Increased complexity for management and decision-making | Complex | 8 | | | |
| | Cultural barriers, confrontation with farming practices of parent's generation | Trad | 7 | | | |
| | Cognitive frame and ways of thinking need to be changed | Cogni | 6 | | | 1 |
| | Seeds are hard or expensive to get | Seeds | 5 | | | |
| | Farmers' lack of awareness about issues linked to specialisation | Awar_Farm | 5 | 1 | 0 | |
| | Lack of available or adapted phytosanitary solutions | Phyto | 3 | 1 | 0 | |
| **From harvest to retail** | Volumes are too limited in a given area to be profitably or easily collected | Coll_Vol | 16 | 1 | 0 | |
| | Equipment for cleaning, drying or storing requires investment | Pre_ProInvest | 11 | | | 0 |
| | Equipment for processing requires investment | Process_Invest | 11 | | | |
| | Competition on the global market with crops produced cheaper elsewhere (for processors or retailers) | Compet | 9 | | 1 | 0 |
| | Equipment for separation of crops requires investment | Separ_Invest | 8 | | 1 | 0 |
| | Equipment for processing requires innovation | Process_Innov | 5 | | 1 | |
| | Regulation issues around sanitary, quality and purity aspects | Qualsan | 5 | | | |
| | Equipment for cleaning, drying or storing requires innovation | Pre_ProInnov | 4 | | 1 | 0 |
| | Administrative, fiscal or accounting issues | Admin | 4 | | | 1 |
| | Equipment for separation of crops requires innovation | Separ_Innov | 3 | | 1 | |
| | Traders are reluctant to support solutions which may reduce the inputs they sell | Input | 3 | 1 | | |
| | Dealing with diversification products incurs higher costs | Cost | 3 | | 1 | |
| **Market** | Need to raise consumers' awareness or bad visibility of diversification benefits | Awar_Comm | 17 | 0 | | 1 |
| | Uncertain or unstable market | Uncert_Mark | 14 | 1 | | |
| | No pre-existing or very limited market | Exist_Mark | 13 | 1 | | |
| | Doubts about willingness of consumers to pay more for diversification products | Willing | 9 | 1 | | |
| **Coordination between value chain actors** | No ensured and/or fair sharing of added value between actors | Price | 17 | | 1 | 0 |
| | No ensured or limited volumes to buy/sell products or establish secure contracts | Quant | 12 | | 1 | 0 |
| | Duration of contracts not enough to secure farmers in taking risks and investing | Dura | 10 | 1 | | |
| | Limited or no cooperation between innovative farmers | Orga | 8 | 1 | 0 | |
| | Individualistic mentality and lack of trust between farmers limit collective action | Indiv | 7 | 1 | | |
| | Unbalanced power in bargaining between farmers and traders | Power | 7 | 1 | 0 | |
| | Finding suitable contracts to address issues related to variability in production (flexibility, sharing risks and reducing control costs) | Variab | 7 | | | |
| | Lack of communication between value chain actors | Comm | 6 | | 1 | 1 |
| | No ensured quality of products to be bought, sold or to establish secure contracts | Qual | 4 | | 1 | |
| | No ensured reciprocal benefits in partnership (especially for land arrangements) | Benef | 4 | | | 1 |

n: number of occurrences of the related barrier among the 25 case-studies. W: "Changing from within" ideal-type; O: "Building outside" ideal-type; H: "Playing horizontal" ideal-type. "1" (with colour related to the value chain level) indicates that the presence of the corresponding barrier was visually linked on MCA to the corresponding innovation setting ideal-type (Fig 4). "0" indicates a link between the absence of the barrier and the innovation setting. The grey colour indicates barriers which were not linked to any ideal-type in particular (transversal);

*Common Agricultural Policy (European Union).

51] and Factoextra [52]). Farming strategies, value chains and type of agriculture shaping the food system innovation setting (Table 1) were integrated as supplementary variables in the MCA to analyse their links to groups of barriers.

In our analysis, we examined the first four dimensions (axis 1 to 4) of the MCA, explaining 42% of total variance (S1 Fig). For each 2-axes projection, only the supplementary variables and barriers most linked to the axes were kept in the analysis (cut-off: v-test absolute value higher than 0.8 for one of the dimensions, corresponding to an alpha error of 10%).

Based on the MCA outputs, we visually connected barriers to three "ideal-types" of food system innovation settings (described in Section 3), which were specific combinations of innovation modalities described by the supplementary variables. The number, visual determination and limits of those groups were guided by the qualitative analysis of material collected with the different innovation teams. We use the term "ideal-types" in the sense of Weberian sociology, to describe theoretical constructs created by researchers to emphasise and understand tendencies observed in the complex reality [53]. The goal of ideal-types is thus not to put concrete cases in one specific category but rather to use such conceptual categories as a heuristic tool to discuss reality, assuming that we can better understand a concrete case while exploring the extent to which and the ways in which it diverges from a particular ideal-type or shares the characteristics of different ideal-types (see Part 5). All statistical analyses were carried out on R (Version 3.5.1). A detailed summary and plots of the different steps of statistical analyses can be found in the supporting information.

## 3 Results

In Part 3.1 we provide a first general description of barriers to crop diversification highlighted in this study. The 3 ideal-types of food system innovation settings: ("Changing from within", "Building outside", "Playing horizontal") and the barriers they are specifically linked to are detailed in Part 3.2.

### 3.1 A diversity of barriers to crop diversification at all value chain levels

A wide variety of barriers to crop diversification (46) is highlighted across the 25 cases, at all levels of value chains from farming to marketing (Table 2). Cases mention in average 16±4 barriers ("±" stands for standard deviation throughout the paper): 7±2 at the production level, 3±2 for operations from farm to retail, 3±2 around coordination issues and 2±1 related to markets.

Among the 25 cases, major barriers to crop diversification at the farm level are related to the lack of technical knowledge and references regarding minor crops and crop diversification, the absence of suitable equipment on the farm, the lack of crop varieties adapted to the local context, fears of increased complexity and uncertainties requiring cognitive changes in farmers' for decision making, and public regulations which do not create strong incentives to diversify.

Downstream of value chains, logistics is a major issue with high costs and no suitable facilities to collect, store and manage small volumes of new crops for actors who historically targeted economies of scale while focusing on large volumes of a reduced number of crops. New crops often compete on the global market with other or similar crops that can be produced cheaper elsewhere. Lack of coordination and designing suitable contracts between chain actors are also highlighted as a real challenge. The necessity of finding or building fair, stable and sufficiently large markets for new products is likewise mentioned as a major barrier. Many doubts are expressed about consumers' willingness to pay more for new differentiated products, so that initial higher transaction costs can be covered to start up new value chains and stimulate

production and interest. A majority of cases highlight difficulties in making benefits of diversification crops visible and understandable to consumers. They underline the fact that the lack of consumers' awareness around issues related to crop diversification and developing adequate communication are central challenges.

## 3.2 Links between barriers and ideal-types of food system innovation settings

Some barriers are not specifically linked to innovation settings, for example the lack of technical knowledge and references for implementing new crops that can be considered to exist across cases (Table 2, Fig 4). We were nevertheless able to link 37 barriers more specifically to 3 ideal-types of food system innovation setting (Fig 4), detailed below. In the text, "1" following the barrier code denotes its presence and "0" its absence.

**3.2.1 Changing from within.**   This ideal-type relates to an innovation setting within the dominant food systems, corresponding to: (i) temporal diversification, which is the most common strategy of crop diversification; (ii) targeting commodity markets; and (iii) including mostly conventional farmers.

A concrete example could be the integration of chickpea or soy in conventional rotations for export in large volumes to the Middle East, to be processed there as hummus for mass consumption or cultivated for the feed industry. 15 barriers were linked to that ideal-type.

In such cases, farmers are generally not well aware or convinced of sustainability issues around simplified farming systems, or of the benefits they could derive from crop diversification (Awar_Farm_1). It is difficult for them to perceive the advantages of adopting crop diversification because their situation is still profitable in the short-term in the current economic and regulatory context, especially given the low prices of fertilisers and pesticides, which are not an incentive to move towards systems with less dependency on external inputs (Current_1; Reg_1; Prof_0). To convince them to switch to more agroecological systems, there needs to be more available knowledge and references proving the potential benefits of crop diversification for the sustainability of their farm (K_Sustain_1).

There is also found to be a lack of knowledge and references about the impacts of integrating new crops or practices at the farming system level (K_Syst_1). For some conventional farmers, it is quite challenging to develop systemic thinking of longer rotations, and to integrate into their decision making the idea that one new crop could have positive impacts for several years. This could be explained by an approach to farming oriented primarily to short-term profits and quick reactivity to commodity markets. This lack of systemic long-term thinking is reinforced by the fact that most agricultural advice given to conventional farmers is focused on short-term production, with references to annual gross margins at the crop level and no accounting for the multi-year effects of crop diversification. (Advice_1). At the farm level, conventional farmers also raise questions about the fact that for some new crops, no satisfactory crop protection solutions based on pesticides exist (Phyto_1). Conventional farmers in commodity value-chains often receive most of their agricultural advice either from cooperatives or from traders who both buy their crops and sell them fertilisers and pesticides. As crop diversification could reduce the dependency on external inputs and the production volumes for main crops, these players have no interest in promoting it (Input_1).

Compared to the "Building outside" ideal-type (below), barriers related to farm machinery are presented here more as a question of innovation than as a problem of investment. (Machin_Innov_1; Machin_Invest_0). This may be connected to the size of farms which is in average higher in cases involving conventional farmers (288±618 ha) than in those involving only organic farmers (50±51 ha).

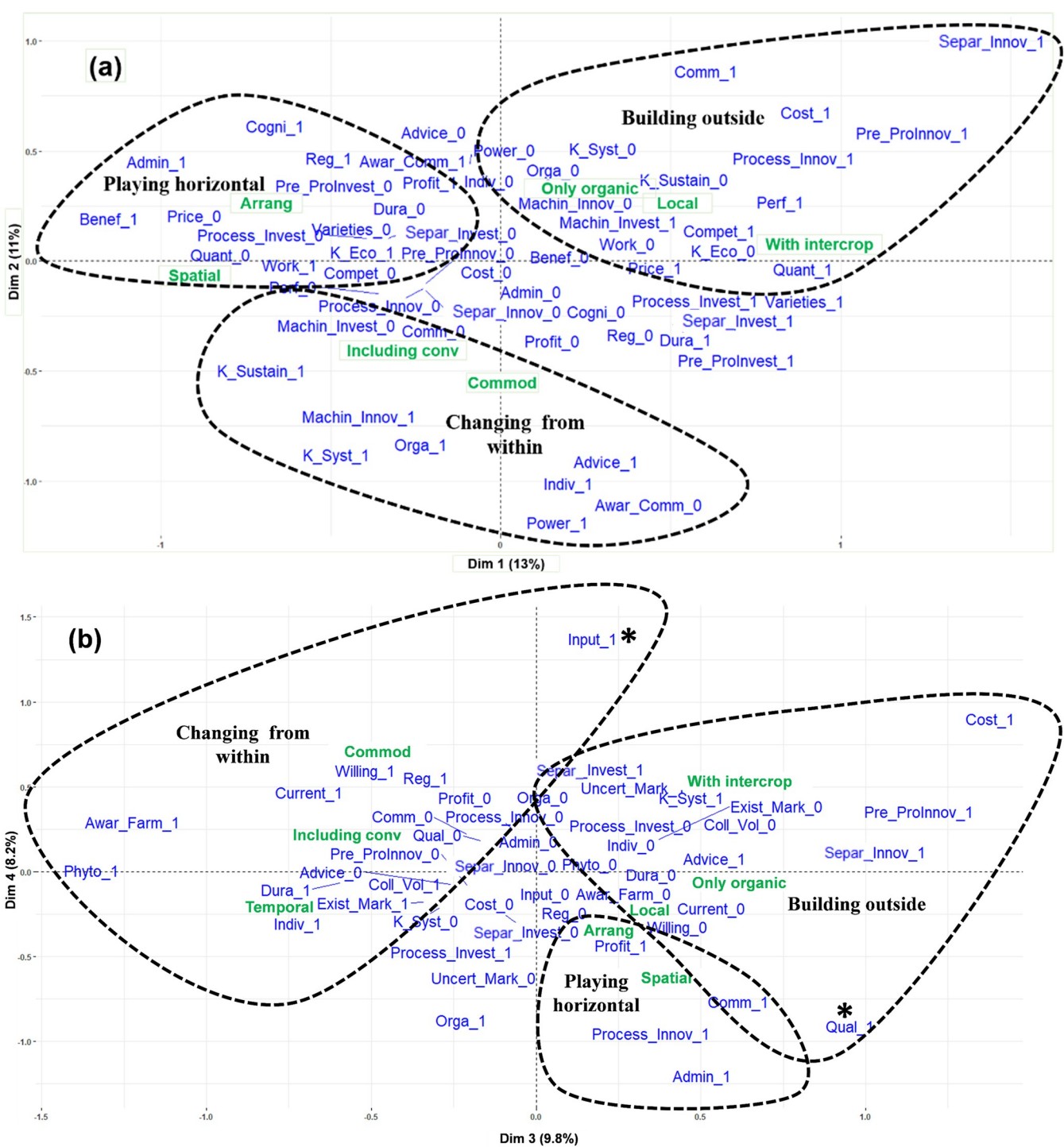

**Fig 4. MCA projection of barriers (in blue) and supplementary qualitative variables related to the food system innovation setting (green) on axes 1–2 (a) and 3–4 (b).** The codes used for supplementary variables are presented in Table 1 and for barriers in Table 2. For barriers, "1" denotes presence and "0" absence. On each 2-axis projection are presented only the barriers and supplementary variables linked to at least one axis. Barriers and supplementary variables were grouped visually into three ideal-types, as shown on the Figure. Although informed by the MCA, the limits of each group were drawn based on the qualitative analysis of interviews carried out with the different innovation teams. This sometimes led to the integration of barriers that were quite distant to the rest of the group (*). Ideal-types and their corresponding barriers are summarised in Table 2.

For operations after production, the major bottleneck is that mainstream cooperatives or traders working with farmers are generally quite big and highly specialised. This makes them reluctant to collect and store small volumes of new crops that innovative farmers may test or seek to develop (Coll_Vol_1). The bargaining capacity of individual farmers against large-scale downstream operators is limited by the fact that these farmers do not easily cooperate to provide together enough volumes to mitigate collection and management costs (Power_1; Orga_1). Innovation teams mentioned that this lack of coordination could be linked to a lack of trust and/or an individualistic mentality frequently found in a competitive environment (Indiv_1). It also limits the possibility to "put some pressure" on traders to convince them that diversification could be a promising strategy.

Guaranteeing a secure outlet in the future is crucial to encourage farmers to "invest in a new crop", as already highlighted in literature [54]. Farmers point out that the difficulty to build multi-year contracts can impede the integration of a new crop in the rotation, which requires financial and personal investment (information, energy, time).

The willingness of consumers to pay more for diversification products raises doubt (Willing_1; Exist_Market_1), either because: (i) actors of commodity value chains prefer to focus on large volumes and low prices to "feed the world"; or (ii) they think that consumers looking for healthier or more ecological products will choose organic products and not conventional ones. The necessity to raise consumers' awareness or to communicate on the potential benefits of conventional crop diversification products is generally not mentioned (Awar_Comm_0).

As innovations are designed within the dominant regime, they are directly bound by the lock-in situation of that regime. Actors at production level seem to have so deeply integrated the constraints and rules of downstream value chains that they barely challenge them. They aim to develop innovations that can be compatible with or adapted to existing mainstream infrastructures and norms, which shows that the highest number of barriers in that ideal-type are related to the production level (Table 2).

**3.2.2 Building outside.**    This ideal-type relates to an innovation setting outside the dominant food systems, corresponding to: (i) integration of intercropping, which is an alternative strategy of crop diversification challenging the existing regime based on single cropping; (ii) developing local markets essentially with small-scale value-chain actors; and (iii) including only organic farmers (Fig 4, Table 2).

A concrete example is the integration on organic farms of very minor crops, pure or intercropped, that are processed into innovative products, e.g. roasted fava bean snacks, lupine cheese or buckwheat pop-corn, by a small-scale food business promoting local food consumption, which contracts directly with farmers for specific volumes of crops and takes in charge the collection, processing and sale of products through online shipping. 13 barriers were linked to that ideal-type.

Contrary to the previous ideal-type, this innovation setting does not seek to fit into the dominant regime but to develop new value chains. The majority of barriers here are thus related primarily to operations after production and the building of these new value chains. The only specific barriers observed at the farm level are the need to invest in farming machinery for specific cropping operations such as weeding new crops, sowing and/or harvesting intercrops (Machin_Invest_1), and the low agronomic performances (especially yields) of minor crops grown in organic conditions (Perf_1). Contrary to innovations involving conventional farmers, the lack of pesticides for managing new crops is not mentioned here (Phyto_0). As noted above, as organic farmers are generally smaller and less equipped than conventional farmers, they need to invest more in such new machines. To increase or stabilise the yields of minor crops cultivated organically, farmers look for technical references (K_Tec_1), just as conventional farmers do (transversal barrier). However, a lack of knowledge about the impacts

of diversification on farm economics, sustainability and the global cropping system is not mentioned specifically as an issue (K_Eco_0; K_Sustain_0; K_Syst_0). Farmers involved in alternative approaches are indeed generally well aware and convinced of the benefits of crop diversification (Awar_farm_0). They are also more familiar with systemic thinking of more complex rotations which is central in organic farming [55]. However, systemic references are still needed on the integration of intercropping in rotations (K_Syst_1).

Farmers innovating outside the regime are generally more willing to collaborate to collectively provide large enough volumes to collectors (Orga_0; Coll_Vol_0), and small-scale operators of local value chains are more used to dealing with small volumes. Interactions between value-chain actors are generally less conflictual, with fewer power imbalances, because most actors are small and share common alternative values (Power_0). In many cases, collection, processing and retailing activities are managed by a small number of intermediaries (short supply chains). However, as the actors are small, marginal, often scattered and not involved in large structured value chains, communication and coordination between them is a challenge (Comm_1).

For such businesses oriented to local niche markets, post-management of crops (especially drying, storing, separation of intercrops, etc.) and processing require investment and innovations, especially for the adaptation of processes to small scales (Separ_Innov_1; Separ_Invest_1; Pre_ProInnov_1; Process_Innov_1). They work with small volumes of products which are generally atypical and more variable in quality than those found in commodity value chains. Altogether, this results in high operating costs (Cost_1). Added to the above-mentioned fact that yields are usually not high, it is very challenging for actors, especially farmers, to make a profit. All the stakeholders underline the necessity of establishing clear contracts in advance, based on fair and transparent prices, and on quantity and quality criteria that cover production and operational costs at all levels of value-chains (Price_1; Quant_1; Qual_1).

Compared to the "Changing from within" ideal-type, actors here doubt less that some consumers would be prepared to pay more for diversification products (Willing_0) if the benefits are clearly indicated with regard to ecology, health, taste or locality. However, although locality is an argument to differentiate products and mitigate costs, competition with non-local crops is still a central issue, even for high-quality or organic products (Compet_1). For example, the development of Swiss organic rapeseed oil is limited by the import of cheaper organic oil produced in countries where production costs are lower (especially manpower), such as Germany or Eastern Europe. So far, the existing literature has underlined competition as a factor limiting diversification of commodity crops, especially in the composition of feed products where the price of substitutable ingredients on the global market is a dominant criterion [16,31]. This work highlights the fact that competition can also be a barrier to diversification of food products in local quality-oriented markets.

**3.2.3 Playing horizontal.** Similar barriers, spread between farm level and other levels of value chains, were addressed by strategies of: (i) spatial crop diversification; and (ii) arrangements between farmers to support new crops (Fig 4, Table 2). Spatial crop diversification (excluding intercropping) can be considered as a step forward to agroecological farming systems compared to only temporal diversification. It is however designed to fit the dominant regime, either through not requiring specific investment in harvesting, or through the separation of crops by actors used to operating on pure products (strip cropping) or to using crop mixes, either: (i) as such on the farm (winter cover crops to enhance fertility) or (ii) sold as fodder to livestock farmers. Arrangements between farmers for selling crops between each other or land exchanges can also be considered as a step forward to more agroecological systems, as far as collaboration between territorial actors and local cycling of nutrients are concerned [56]. We grouped these two strategies, which can be complementary, into a single ideal-type that we called "Playing horizontal". We chose that name as actors promote quite radical alternative

strategies, either strictly speaking horizontal at spatial level (crop mixes and strips) or socially horizontal (arrangement between farmers), without directly challenging the vertical organisation of dominant regime value chains. 9 barriers were linked to that ideal-type.

At the production level, farmers in this innovation setting need economic references on diversification practices (K_Eco_1) because they doubt their profitability (Profit_1). For example, an arable farmer needs to be convinced that letting the neighbour's sheep graze his winter cover crops would bring fertility to his plot (reducing fertilisation costs) and that the sheep will totally destroy the winter crops before he sows a new crop in spring, to be sure to save money on that operation. In cases of strip cropping, farmers also wonder about the economic impacts of such practices which are not as well documented as intercropping. Growing strips is supposed to reduce the quick spread of diseases that occurs in homogeneous fields and is facilitated by interactions between crops at the interface of strips [40]. Compared to intercropping, strip cropping is also supposed to reduce investment in machinery, as regular machinery can be used on each strip which is managed as a single crop (Machin_Invest_0). It does not either require crop separation equipment (Separ_Invest_0). The literature is sparse when it comes to the ideal size of strips for each type of crop, and the expected benefits or potential drawbacks and competition between strips. In any case, managing strips or collaborating with livestock farmers requires a shift in the way of thinking about interactions, either at the plot/farm level or between farms (Cogni_1). Such changes raise concerns about labour organisation and mental load (Work_1), e.g. if arable farmers need to host sheep in winter or if strip cropping farmers have to operate on a similar crop on strips which are now scattered.

Regulations and administrative issues are also a strong barrier (Reg_1; Admin_1). Contracting directly between farmers for selling crops is for example highly complex in some countries from a tax perspective, especially for big volumes. It is sometimes required to be supported by an often reluctant intermediary (such as a cooperative) which is officially allowed to collect taxes for the state, whereas farmers would like to interact directly between themselves. The official declaration of strips in dedicated software to obtain public subsidies (CAP) is a brain teaser, because such software usually allows only one crop per field. In the case of land exchange between farmers, questions are raised about who will get the subsidies attached to each plot.

As the selling and processing of crops either fit the dominant regime or are managed at farm scale, post-harvest operations and competition for selling the crops are not mentioned as limiting factors (Pre_ProInnov_0; Pre_ProInvest_0; Compet_0). As farmers deal directly with one another or with their usual trade partners, contracts are not an issue as far as prices, quantities and duration are concerned (Price_0; Quant_0; Dura_0). However, in arrangements between farmers, the lack of communication between them can be an issue, especially to clearly objectify the advantages and compensation expected by each of them (Comm_1). Emphasis is therefore put on the need to draw up fair contracts guaranteeing that both livestock and arable farmers will derive benefits, especially in cases of grazing on winter cover crops and land exchanges (Benef_1). In the case of land exchange, farmers often wonder about fair "exchange ratios" and whether they can prove that one piece of land is better than another one, or if the advantage expected for one farmer is higher than for the other.

As regards the promotion of their crop diversification strategies, farmers underline the difficulties in communicating to consumers (Awar_Comm_1) because their farming practices are quite hard to understand if one has no farming background. It is for example a challenge to explain to consumers why crops grown in strips rather than in field would be more beneficial to the environment. This is why farmers who practice strip cropping prefer to communicate on the positive visual aspects of regular strips on the landscape. Similarly, communicating on land exchange to differentiate products is not easy (how can one explain simply that a type

of flour is better because the wheat was grown on a plot where a livestock farmer had previously grown grass for grazing?).

## 4 Discussion

### 4.1 Methodological limits and possible improvements

The building of ideal-types was based on MCA carried out on 25 innovation cases involved in the DiverIMPACTS project, which were not initially selected to build an optimal design of experiments but rather to cover a wide diversity of situations. Although we ensured that every modality of supplementary variables presented at least five occurrences, not all possible combinations of variables were present in balanced proportions. The less frequent barriers occurred only three times across the 25 cases. The links between the less frequent barriers (at the bottom for each value chain level in Table 2) and innovation settings therefore need to be interpreted carefully. Although some well-represented barriers were linked specifically to one ideal-type, we must bear in mind that ideal-types are heuristic constructs simplifying reality, and that barriers were often not exclusively for one given combination of variables describing innovations settings.

The visual "drawing" of ideal-type boundaries and connected barriers on MCA outputs (Fig 4) was subject to our human interpretation to "make sense" of data (informed by the qualitative analysis of material collected with innovation teams). This exploratory approach could be enhanced by more systematic methods of clustering on a larger dataset. The impact of less frequent barriers on ideal-types could also be tested while comparing MCA outputs integrating only more frequent barriers.

In our analysis, the first four dimensions (axis 1 to 4) of the MCA explain only 42% of total variance. This suggests that although innovation settings play a role in shaping the barriers to crop diversification, other factors should be considered. Studying more cases would allow us to strengthen the robustness of the analysis and also to integrate new variables that we did not integrate because of the small sample. For example, a preliminary online survey carried out in the DiverIMPACTS project (with 129 answers) [57] showed that the geographic zone could impact challenges and drivers to crop diversification. In our study, conventional farms were in average bigger than organic farms (see 3.2.1). Integrating the size of farms as a separate variable could be relevant as organisational, material and financial sustainability challenges are likely to be connected [58]. We chose not to integrate the size of farms and the country (11 possibilities) in our analysis because the possible modalities of such variables were not well balanced across the cases.

The type of crops used for diversification can also determine some barriers, as highlighted in the literature [16, 31]. In our sample, we could not integrate crop type as a variable because most cases involved a diversity of crops.

### 4.2 Farmers' perspective, agency and innovation dynamics

The analysis of barriers to crop diversification was based on interviews with innovation teams' leaders and monitors who mostly belonged to farming-related organisations (farmer's associations, agricultural R&D, extension or advisory services). The ideal-types that we built thus account for the perception of barriers along the value chain from farmers' and their support services' perspective, rather than providing an integrated analysis based on the expertise and points of view of actors at different value chain levels, as found in other studies [14–16,31,36–38]. In the methodological section (Part 2.2) we have already mentioned that the precision in the description of barriers was impacted by the fact that innovation teams were farm-oriented. To ensure a similar level of precision in the characterisation of barriers at all value-chain levels,

which may for example result in creating more detailed categories for barriers after the farm gate, we could hold complementary interviews with other value-chain stakeholders, which may have more expertise on those aspects.

Although this creates a bias, it may inform us about the level of agency, defined as the capacity to take purposeful actions in an attempt to generate change [59, 60], perceived by farmers to act on the food systems. In the "Changing from within" ideal-type, farmers involved in conventional commodity value chains focus their scope of action at the farm level, accepting the rules of downstream actors, or feeling powerless to change them. Conversely, farmers of the "Building outside" ideal-type have decided not to accept those rules and to collaborate with other small-scale businesses to create value chains with new rules. In that sense, their level of agency seems broader. The "Playing horizontal" ideal-type may denote an intermediary level of agency regarding both farming practices and collective action at the territorial scale between farmers, without extending to downstream operations. Interviews with innovation teams were carried out in the starting phase of the DiverIMPACTS programme. In that sense, our work provides a static initial diagnosis of the challenges faced by innovation teams. Innovation and transition are dynamic processes [24, 61–63]. This programme thus aims at gradually nudging actors out of their comfort zone, empowering them and stimulating co-innovation between multiple actors at different value chain levels. Further research should explore the extent to which this first overview of challenges will determine innovative actions undertaken, and how actors' perception of barriers at different levels of food systems change during the process, enriched by other actors' points of view or new alliances, modified and informed by action in a logic of reflexive interactive design [64, 65].

The role that researchers can play in innovation dynamics at the value-chain level should also be explored more fully. During the workshops, we (as scientists) provided guidelines to innovation teams to support them in identifying the different barriers they face (S1 Appendix). This framework was based on a preliminary literature review and underlined the necessity to consider multiple dimensions of barriers at all levels of the value chain. It also provided a first list of potential barriers to explore with stakeholders. This framework may have helped to broaden innovation teams' perspective beyond the aspects they would normally consider in their professional activities (mainly dealing with farm-level issues). On the other hand, it may have determined the ways in which innovation teams interacted with other value-chain actors, and could have limited the exploration of dimensions not previously mentioned in the literature. In this regard, it seems important for scientists to develop reflexivity in exploring acceptable trade-offs between: (i) providing scientific information to accelerate/broaden change processes with a risk of pre-defining innovation pathways too much; and (ii) playing only a role of facilitators that may allow more original development but does not enable value chain actors to benefit from previous scientific knowledge.

## 4.3 Supporting innovation according to food system settings and patterns of challenges

The main added value of our work is to provide a comparative analysis of barriers to different food system innovation dynamics that have previously been investigated in distinct studies either on: (i) lock-in situations preventing sustainability innovations in dominant food systems [12–16, 31, 35–38, 66]; (ii) impeding factors and success conditions for alternative innovation dynamics (born outside the dominant system) to be developed [1, 9, 22, 24, 26–28]; or (iii) challenges to support crop-livestock integration at the territorial scale, especially for the "Playing horizontal" ideal-type [42, 43, 56]. Most of those studies investigated in detail a limited number of cases in specific countries. Our study did not seek to examine each innovation case

in depth, but rather to systematically investigate a diversity of European innovation settings. It confirms the prevalence of impeding factors to agroecological innovation identified in those three literature streams across a diversity of European contexts.

The comparative dimension of our work across a diversity of food systems allows us to show that diverse innovation settings of agricultural practices and value chains can result in contrasting patterns in the combination of barriers. This diversity of challenges has to be taken into consideration to develop targeted research, innovation and policy actions in relation to the food systems they seek to support. We formulate specific recommendations in that sense (Fig 5) but think that both the characterisation of food system innovation settings (e.g. integrating more detailed variables) and the corresponding adapted action priorities deserve further research. Those specific R&D and policy actions could be articulated with ones targeting barriers that are not linked to one innovation setting in particular and can therefore be considered as more generic. This is the case of barriers outside of ideal-types boundaries (Fig 4), shared across two innovation settings or not specifically linked to any of them (Table 2). For example, such transversal interventions could foster the development of technical references for growing new crops (K_Tec) and the access to seeds and varieties of minor crops adapted to a diversity of local conditions (Varieties, Seeds). They could also support investment in post-harvest and processing facilities (Pre_ProInvest, Process_Invest), and encourage suitable contracts between value-chain actors to share the risks associated with the variability of production, especially during the first years when farmers are experimenting with new practices (Variab).

In this study we have characterised each barrier in a binary way (present or absent). To inform policy planning and innovation strategies, a deeper understanding of barriers seems required, especially of their relative "limiting power" (totally blocking, partially limiting, etc.), and of the possibility/necessity to remove it or to adapt innovations if they are not likely to be solved.

| **Ideal-types of food system innovation setting** | **Main barriers to crop diversification** | **Specific priorities to support crop diversification** |
|---|---|---|
| 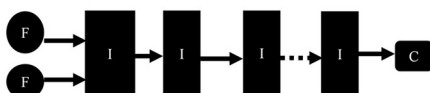<br>**Changing from within**<br>Conventional farming practices promoting longer rotations (temporal diversification), big farms, long commodity value chains with many large-scale agroindustrial and retailing intermediaries. | Profitable current situation; power unbalance between the agroindustry and loosely coordinated farmers constraining innovation at the farm scale to fit to existing downstream infrastructures and requirements based on large volumes and standardised procedures; cultural inertia and complexity of integrating systemic long term thinking in short-term profit-oriented simplified farming systems; belief that technological innovation (machinery) or new phytosanitary products are compulsory to allow change; doubts in the possibility to differentiate new crops based on ecological or health benefits for conventional commodity crops. | Farm and field research on longer rotations in conventional farming with systemic long-term assessment of benefits and relevant management tools for decision making to convince farmers.<br><br>Public subsidies to support diversification crops to be grown on enough acreage in order to encourage change and investment of agroindustry in new crops with economy of scale.<br><br>Supporting collective organisation of farmers to decrease power asymetries and ease negociation. Legitimating crop diversification with arguments related to profitability and image benefits to convince the agroindustry to change [33]. |
| 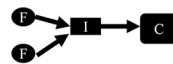<br>**Building outside**<br>Organic farming practices integrating minor crops and intercropping, small farms, short local value chains involving a limited number of small-scale downstream actors. | Actors share strong values and are convinced of ecological practices but struggle in developing new value chains, securing profitability and efficient coordination because of high transaction costs, low and/or uncertain yields/quality of minor crops in organic conditions. Although alternative actors are confident in the possibility to differentiate new local products based on sustainability criteria, doubts remain as far as fair prices are concerned because competition exist with organic products produced cheaper in other conditions. | Farm and field research on organic minor crops and intercropping.<br><br>Research and innovation to develop coordination, post-harvest management/processing technologies, contracts, logistics and organisational modes adapted to small actors and volumes. Public subsidies to support investment for small actors.<br><br>Collection and analysis of success factors for small-scale value chains, about which information is scattered. Creation of a European network of small-scale alternative food chain actors to share experience and organise political lobbying ? |
| 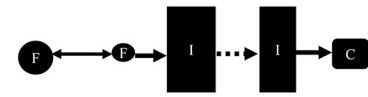<br>**Playing horizontal**<br>Crop diversification supported by territorial arrangements between arable and livestock farmers and/or spatial diversification without harvested intercrops. No change in downstream value chains. | Uncertainties and lack of knowledge on the sustainability and profitability of innovative practices that have been poorly documented; doubts about the possibility to secure partnerships between farmers and ensure fair mutual benefits in the case of territorial arrangements ; cognitive and administrative frames not adapted to innovative practices at new scales (strips, territorial collaboration); difficulties in communicating to average consumers (with no agricultural background) on specific implemented practices, apart from the positive impact on landscapes. | Farm and field research on strip cropping, land exchange, winter grazing of arable mixes of cover crops and development of relevant management tools for decision making.<br><br>Evolution in regulations and CAP declaration softwares to ease administrative and accountancy challenges (e.g. tax collection) faced by farmers willing to cooperate at territorial level (land exchange, direct sale of crops between farmers) or growing strips.<br><br>Designing suitable farmer-to-farmer contracts to ensure mutual benefits of territorial collaboration. |

**Fig 5. Main barriers and priorities to support crop diversification according to ideal-types of food system innovation setting.** "F": farmer"; "I": intermediary actor of value chain (collector, processor, trader, retailer); "C": consumer. Arrows represent the flows of products from production to consumption along the value chain. The size of value chain actors is in line with the average size of actors in the different ideal-types, which tended to be smaller in local and organic value chains.

### 4.4 Analysing hybrid and intermediate food system innovation settings

Our conceptual proposal of food system innovation settings is based on 3 criteria with 3 modalities, which implies 27 combinations. Of the 27 possible combinations, 16 of the 25 case studies (64% of cases) are in 4 of the 27 possible combinations (14.8% of the possible combinations).

This study was based on empirical data collected on the 25 cases of the DiverIMPACTS project which were not initially selected to cover all possible combinations of innovation settings. In those cases, there is a very strong link between local and organic chains, and between conventional farming and commodity value chains. This can be explained by the fact that organic farmers may be more likely to be involved in local value chains. Organic agriculture was developed as an alternative to dominant agriculture, and promoting shorter supply chains has historically been a strong strategy to support the development of that innovation niche [10, 22]. The resulting imbalance in possible combinations of modalities in our sample implies that in our analysis we are not allowed to distinguish the relative contribution of "organic" and "local" in the "Building outside" ideal-type, and of "conventional" and "commodity" in the "Changing from within outside" ideal-type. In that sense, those two food system ideal-types echo the dichotomy of radical *versus* incremental [1, 18, 23], transforming *versus* conforming [25, 67], and alternative *versus* dominant [10, 68, 69] found in literature. The third ideal-type that we develop, "Playing horizontal", tends to show that "things may be more complicated out there" and that even as a heuristic tool, binary categorisation may be limited when trying to grasp food systems' complexity.

Recent studies show moreover that hybrid foods systems are developing, e.g. combining for example conventional and local chains [71], or organic and commodity chains [55], as in 3 of our 25 cases (Table 1). Would the development of organic intercropping be facilitatecd or complicated by the integration in well-structured dominant value chains rather than local ones? Would new crops in conventional rotations face fewer or more problems if they targeted local quality markets instead of export as commodities? To disentangle the specific contribution of agriculture type and value chains in such hybrid settings, and potentially create more complex ideal-types, our methodological approach could be applied to a wider sample of cases that ensures a balanced cover of all possible combinations.

Industrialised countries are witnessing the growth of intermediate food value chains with structured industrial-like organisation but a strong connection to local farmers and farmers' groups, with constraining traceability, fair share of added value, and ecological practices. The potential of combining the benefits of long and short supply chains in such hybrid food systems is starting to be explored under the umbrella concepts of "agriculture of the middle", "mid-tier supply chains" or "values-based food supply chains" [70, 71]. To analyse the specific barriers to diversification in such hybrid chains, we could integrate new possible modalities in our approach, such as "intermediate value chains". Another possible research approach would be to discuss with actors involved in hybrid value chains the similarities and differences in the barriers they face compared to our "pure" ideal-types. Doing so, we may be able to highlight whether: (i) hybrid configurations face cumulated challenges of the "Changing from within" and "Building outside" categories; and (ii) hybrid configurations remove or create more barriers.

## 5 Conclusion

Based on the analysis of 25 European cases promoting crop diversification, we propose to characterise food system innovation settings as combinations of farming practices, agriculture type, and value chain. In this study we highlight three ideal-types of food system innovation

settings: (i) "Changing from within" where longer rotations are implemented on conventional farms involved in commodity supply chains; (ii) "Building outside" where crop diversification integrates intercropping on organic farms involved in local supply chains and (iii) "Playing horizontal" where actors promote alternative crop diversification strategies, either strictly speaking horizontal at spatial level (e.g. strip cropping) or socially horizontal (arrangement between farmers) without directly challenging the vertical organisation of dominant value chains.

Each ideal-type is linked to a specific pattern of barriers to crop diversification. For example, in the "Changing from within" ideal-type, farmers aim to develop innovations that can be compatible with existing infrastructures and norms of big agro-industry players. This shows that the highest number of barriers are related to the production level, such as developing knowledge and management tools to integrate new crops in historically simplified and short-term profit-oriented production systems. Conversely, the "Building outside" innovation setting is not intended to fit in the dominant regime. The majority of barriers in that case are related more to the building of new value chains and to post-production operations, such as developing adapted technology for the post-harvest management and processing of new crops on a small scale. In the "Playing horizontal" idea-type, post-harvest operations are not mentioned as limiting factors because the selling and processing of crops either aim to fit the dominant regime, or are managed at farm scale. Barriers here are rather related to changes required in cognitive, regulatory and administrative frames to facilitate spatial innovations at new scales (crop strips on the farm, territorial collaboration between farmers).

This work affords a better understanding of the specific barriers that should be considered to develop targeted research, innovation and policy actions for the food systems they seek to promote. It contributes to recent research developments aspiring to better analyse the diversity of food systems, with a view to supporting transition [2, 39].

## Supporting information

**S1 Appendix. Guidelines and structured framework to support innovation teams in identifying barriers to crop diversification in the DiverIMPACTS project.**
(DOCX)

**S1 Fig. Percentage of inertia explained by the different dimensions of MCA for barriers to crop diversification.** Cumulated, dimensions 1 to 4 explain 42% of variance.
(TIF)

**S2 Fig. MCA ellipse plots of barriers to crop diversification and supplementary variables on dimensions 1 and 2.** Ellipses around the different modalities of each variable indicate significant difference with a confidence of 95%. Codes for barriers are presented in S1 Table and for supplementary variables in Table 1.
(TIF)

**S3 Fig. MCA ellipse plots of barriers to crop diversification and supplementary variables on dimensions 3 and 4.** For legend, see S2 Fig.
(TIF)

**S1 Table. V-test of coordinates of supplementary variables on dimensions 1 to 4 in the MCA of barriers to crop diversification.**
(DOCX)

## Acknowledgments

We are grateful to all the partners of DiverIMPACTS who have inspired us, and to the innovation teams that enthusiastically took part in the research. We especially thank Walter Rossing, Pieter de Wolf, Jorieke Potters, Daniel de Jong (Wageningen University and Research) who organised the co-innovation workshops where our interactions with the innovation teams took place, and the project leader Antoine Messéan (INRA), for his support and advice. We thank the academic editor and the three reviewers for their complementary constructive feedback on our paper.

## Author Contributions

**Conceptualization:** Kevin Morel, Eva Revoyron.

**Data curation:** Kevin Morel.

**Formal analysis:** Kevin Morel, Eva Revoyron, Magali San Cristobal.

**Funding acquisition:** Philippe V. Baret.

**Investigation:** Kevin Morel, Eva Revoyron.

**Methodology:** Eva Revoyron, Magali San Cristobal.

**Project administration:** Philippe V. Baret.

**Software:** Magali San Cristobal.

**Supervision:** Philippe V. Baret.

**Writing – original draft:** Kevin Morel.

**Writing – review & editing:** Kevin Morel, Eva Revoyron, Magali San Cristobal, Philippe V. Baret.

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
