## [Decision Letter · Decision Letter 0]

22 Aug 2019

PONE-D-19-17348

Innovation within or outside dominant food systems? Different challenges for contrasted strategies of crop diversification in Europe

PLOS ONE

Dear Kevin Morel,

Thank you for submitting your manuscript to PLOS ONE. After careful consideration, we feel that it has merit but does not fully meet PLOS ONE’s publication criteria as it currently stands. Therefore, we invite you to submit a revised version of the manuscript that addresses the points raised during the review process.

ACADEMIC EDITOR: Your manuscript has good potential to contribute to the scientific community, but it needs substantial improvement to be accepted by PLOS ONE. Despite differences in the overall evaluation of your manuscript between the three reviewers, all of them ask for a clearer and more explicit description of the research design and methodological approach. Furthermore the structure of the manuscript and specific sections need to be improved, as recommended by reviewer 1 & 3. Finally, you need to respond to the fundamental criticism of reviewer 3 doubting the issue of hybridization between niches and dominant regime being approached thoroughly in your study. Also the "imbalance" in your available case studies, with the large majority dealing with one of 27 possible combinations in your conceptual approach needs to be thoroughly discussed.

We would appreciate receiving your revised manuscript by 21 September 2019. To enhance the reproducibility of your results, we recommend that if applicable you deposit your laboratory protocols in protocols.io, where a protocol can be assigned its own identifier (DOI) such that it can be cited independently in the future. For instructions see: http://journals.plos.org/plosone/s/submission-guidelines#loc-laboratory-protocols

We look forward to receiving your revised manuscript.

Kind regards,

Til Feike, PhD

Academic Editor

PLOS ONE

Journal Requirements:

2. Our internal editors have looked over your manuscript and determined that it is within the scope of our Future Crops Call for Papers. This collection of papers is headed by a team of Guest Editors for PLOS ONE. The Collection will encompass a diverse range of research articles on enhanced agronomic production, guaranteeing food security and neglected crop species.  Additional information can be found on our announcement page: https://collections.plos.org/s/future-crops.

If you would like your manuscript to be considered for this collection, please let us know in your cover letter and we will ensure that your paper is treated as if you were responding to this call. If you would prefer to remove your manuscript from collection consideration, please specify this in the cover letter.

3. We note that Figure 2 includes an image of a patient / participant in the study.

Reviewers' comments:

Reviewer's Responses to Questions

**Comments to the Author**

1. Is the manuscript technically sound, and do the data support the conclusions?

Reviewer #1: Yes

Reviewer #2: Yes

Reviewer #3: Partly

2. Has the statistical analysis been performed appropriately and rigorously? 

Reviewer #1: Yes

Reviewer #2: Yes

Reviewer #3: Yes

3. Have the authors made all data underlying the findings in their manuscript fully available?

Reviewer #1: Yes

Reviewer #2: Yes

Reviewer #3: Yes

4. Is the manuscript presented in an intelligible fashion and written in standard English?

Reviewer #1: Yes

Reviewer #2: Yes

Reviewer #3: Yes

5. Review Comments to the Author

Reviewer #1: The authors question the paradigm and dichotomy of aiming at innovations either within the dominant farming systems or outside those. The authors use crop diversification as a case for agroecological innovation and sample barriers towards adopting different forms of it. As the study is not only about innovation systems, but also explicitly about where these are embedded in, the authors introduce the more comprehensive concept of “food system innovation settings”. In this way, they draw attention to the manifold frame conditions of innovations and to the complexity inherent to each innovation system as such. By doing so, the authors build evidence for the targeting of policies aimed at effectively supporting change. In particular, the comparative perspective on different farming styles (conventional, organic) in a sufficiently detailed qualitative analysis makes the manuscript an enjoyable piece of work.

Overall, it is an interesting and solid piece of work. The authors have carefully handled their results. The methods chapter needs some improvement as does the discussion. In general, the manuscript is quite readable. The authors require to check some format and language issues.

More details:

One part of the methods needs more information. The case selection apparently simply follows a project rationale. This is per se acceptable but some explanation why the project is/was working on such cases would be needed – and coming to this point just in the discussion is far too late. What is even more required is some more insight into the type of data collected. Was it mere observation during the course of the project? or was there any maybe structured assessment of information from each “innovation team”? Suddenly, you mention “interview content” (L202) – what you did with whom needs to be clearly outlined. Figure 2 is helpful, though not enough as a stand-alone. Even the discussion (far too late) suddenly reveals some more details about the methods (L488-9). Comprehensive information about data collection is most relevant when reading the results chapter, where at the beginning of that section it is not always clear whether you are reporting results or are already discussing those. However, this remark does not apply to the sections 3.2.1-3, where the results are nicely referenced and clear.

There are few incidences with choice of terms. For instance, L240 “factors are correlated” – the choice of the word correlated suggests that it is a matter of statistics, what it is apparently not at this point. More confusingly, this incidence finally is from a reference, while it is placed in the beginning of the results section.

While the informative sub-titles guide well through the results, the sequence of presenting the results is not always clear to me. For instance, in 3.2. you start with the markets – why about markets first? (L259: not Tab 1 but Tab 2 meant?) Another example, in Table 2, you already introduce the three “ideal types”, but how you arrived at those comes later. I think you should have presented the MCA earlier. L266-268: this is discussion, not results. The first section of 3.2 (before starting with 3.2.1) needs thorough revision, or complete rephrasing. I would introduce the MCA result here instead of just mentioning them casually in L278. The Power barrier is quite discriminating in axis 1, Quant in axis 2, etc., a lot of zero-labeled codes in the middle.

What finally remains open about the ideal types is why you decided to have three ideal types and not four, or any other number. The remainder of the non-associated data points also seems to cluster together. How did you form the ideal types? There is some information given in Fig 3, but the text is incomplete on that issue. It is good to avoid repetition in a manuscript, but the text needs to be comprehensible also without the notes in the tables and figures.

Moreover, it is remarkable that the four first axes of the MCA only yield about 40% of the total variance. Similarly, the formed clusters are not obvious regarding the layout of the data points. This requires some critical discussion.

L482-5 makes an outlook to methodological development. I think it is more than that, it is about the usefulness and applicability of the results, the next steps when aiming to transform the scientific results into planning action, either in a scientific or policy attempt. It should be further discussed in this sense. L540ff appears to be somewhat connected to the same issue. You shortly mention recommendations and action priorities that are shown in (the hardly readable) Fig 4.

The authors mention variables that could have further sustained their analysis, such as farm size. One factor that is already recorded is the country. Surprisingly, the authors mention some differences but do not do so in a systematic manner. It remains unclear why that factor has not been further followed up.

The conclusions read very well. There is only one point that needs attention: power asymmetries and games. They come a bit out of the blue and are then prominent in your final statement. Please better connect to your data, maybe even need to present additional data in the results section.

Overall, the manuscript is quite readable. Nevertheless, there are still some linguistic challenges regarding the employ of terms and punctuation, and to some extent grammar and spaces. Examples: Terms, L108 the crops “are very contrasted”, L71 Rotations are “enhanced”, L129 “screening” of harvested crops. Punctuation, L70 to promote “omega 3 rich products”: many words in a row that could be better structured through hyphens, also L161, also outside niche innovations. Grammar: L183 multi-step procedure (not steps here). These are examples, the incidences are not limited to these. Moreover, the reference list still contains some French and there are several issues with upper/lower case writing. Table 1 is an example of raw data that needs some processing by the authors: some sorting is required – what is the rationale behind the current sequence of rows? In the text, the use of the normal brackets instead of the square ones for indicating reference is somewhat irritating – and even more, when other numbers are also put in such brackets (L230). There are some very long sentences (up to 5 lines), which need structuring through breaking down to smaller sentences. L397-8 lacks a verb – incomplete sentence. Small but intriguing: L585 not processes but processors meant? MLP does not deserve to be abbreviated, it is too much spread over the manuscript and not often used. From what I learned, English sentences never start with a figure – always spell out numbers at the beginning of a sentence. The font size of the figures, in particular number 4, is quite small and probably too small. The figure 4, as reproduced for the review, is hardly readable; what is the meaning of the different arrow types and the different sizes of the elements? Table 2: is the color in the right columns the same as a “1” in the codes? If so, then better use “1” in both cases. In general, the authors have made considerable efforts to provide comprehensive legends, which is highly appreciated.

Reviewer #2: The authors aim at presenting an evidence base for analysing innovations in the food systems fostering crop diversification. The central claim is that by identifying patterns of barriers of innovation along the value chain in various settings, the authors will point towards better-targeted research and policy actions. The main strength of the paper is that it goes beyond the dichotomist perspective of the food system and develops a typology of food system innovation, taking into account the hybridisation of food systems.

I would suggest the following changes for more clarity for non-specialists:

- start with one single research question and tell why this is interesting to study - how innovation can start and what is its locus (farm, chain or system)?

- add more details on your theoretical standpoint and the research design

- original data should be available in Zenodo or similar repository

- introduce your definition of the food system, food regime - e.g.line 53.

- provide more detail of the case studies and the "barriers" workshops for the curious reader: visuals, verbatims, participatory observation notes or drawings, etc.

- more reflection on self-referentiality: how your role as researchers can change the innovation settings.

Reviewer #3: As it stands, this article requires many improvements in order to be published.

The authors' thesis is that it is necessary to go beyond a dual approach to transition and innovation processes that would distinguish a dominant regime with incremental innovations and niches that support radical innovations. According to them, the literature most often favours an in-depth analysis of one of the two situations without considering the possible hybridizations and interconnections of the two. There is therefore a lack of comparative and large-scale approaches to better understand barriers to innovation in a diversity of contexts that do not strictly speaking rely on a dichotomy between dominant and niche actors. This is therefore what the authors propose to do by addressing the barriers to crop diversification along agri-food systems (production, collection, processing, retail, market and actors coordination). The authors use 25 European case studies as a basis for this. And they introduce the concept of food system innovation settings combining a type of innovative agricultural practice, a type of value chain supporting this innovation and a type of agriculture (organic or not).

My main criticism is that I do not see the added value of their work on the issue of hybridization between the dominant regime and niche actors. The authors point out that this has never been done in the literature, but I am not convinced that they really address the issue of hybridization between a dominant regime and innovative niches. On the theoretical level, however, there is a whole literature that has addressed the question of the articulation between dominant regime and niche, starting with the multi-level perspective, which is also mentioned many times in this article. Also it seems to me that on the one hand the authors minimize the contributions of these studies on hybridization between niches and dominant regime. On the other hand, the authors use a very normative approach, consisting in categorizing the social world they are studying, a categorization that should be discussed and which seems difficult to reconcile with a detailed analysis of the hybridization processes to which the authors aspire. The concept of “food system innovation settings” seems more suited to the variability of the case studies mobilized than to a real contribution as a concept for a better understanding of the barriers to transition to more sustainable systems.

The following points therefore need to be clarified:

The basic proposal for the “food system innovation settings” is based on 3 criteria with 3 modalities, which implies 27 possible combinations. Are these combinations all possible? How could some of them be subject to more or less strong barriers? How are some of them more or less similar or more or less compatible? Authors must discuss these questions if they want to use these criteria in the service of a new concept.

Each criterion is also worth discussing.

Crop diversification practices: what is the scientific justification for these 3 categories? This should be clarified because there seems to be a mix of agronomic and technical criteria. It is in particular the distinction between spatial and intercropping that involves a purely technical criterion, whether or not to screen the harvest, that poses a problem. When is it decided that a technical criterion justifies the creation of a new category? And who decide this?

Sale of products: the authors propose two rather binary categories: commodity with long chains and export and local with short chains and alternative food systems. The introduction of a third category refers to something quite specific related to exchanges between farmers and stockbreeders.

Type of agriculture: the authors distinguish three cases: AB, conventional and both. But why do they distinguish these 3 cases? What hypotheses behind them?

By discussing these questions, it may be easier to understand the value of this grid and how it can be mobilized.

The 25 case studies in the grid: of the 27 possible combinations, 16 of the 25 case studies (64% of cases) are in 4 of the 27 possible combinations (14.8% of the possible combinations). In particular, there is a very strong link between local and organic and between including conv and commodities. A slightly more in-depth analysis is needed to determine why these combinations are more common.

Methodology section

We lack information on the methodology for collecting case study data. In particular, some elements are introduced into the discussion as limitations when they should be presented in the methodological part. In particular, it is necessary to specify:

- How the case studies were selected?

- How the innovation team was set up?

- Who participated in the 2-hour brainstorming session?

- How was the list of barriers drawn up, given that some formulations are very similar to each other?

Finally, the authors present in a discussion section in 4.2 the fact that barriers result from farmers' perceptions and related advisory services. This is indeed a limitation but it must be presented beforehand in the methodology section.

Result section

This section is poorly structured. If the 3 ideo-types constitute the main result, they should be highlighted from the beginning of this section.

General information is introduced before Table 2, including the paragraph just above (lines 242 to 247). To say that logistics is not adapted to small volumes and that new crops are in competition on the global market seems a little dated and not directly related to the results at this stage.

The first 2 paragraphs of 3.2 seem out of scope. The first (lines 258-265) is focused on market levers and the second (lines 266-268) is a reminder of the merit of this work.

The presentation of the 3 ideo-types is clear (lines 274-457).

Discussion section

The discussion seemed more a reminder of the limitations of this work than a real discussion. I think it is possible to go further by questioning these 3 ideo-types in relation to the grid proposed at the beginning.

Section 4.1: The main message you are sending is that your results should be considered as not very robust (lines 466-471) since some barriers only concern a small number of cases. Is it possible to go a little further? In particular, it seems that it is the Building outside strategy that is most affected by barriers based on a small number of cases

Section 4.2: Yes, you need to recall the limitations of barrier identification and how to improve barrier identification. Your analysis of the agency level seemed a little caricatural to me. Also avoid referring to article under review (60).

Section 4.3: you recall that the literature has focused on a type of dominant food system versus alternative food system situation, most often at national levels, but finally what is the added value of your approach compared to existing ones?

Lines 525-535: What you are saying has already been said in the literature

Section conclusion

It is necessary to recall your main results

Why so many questions in the first paragraph (lines 562-567)

There are a lot of references in the conclusion, see in particular the last paragraph.

You argue that your ideo-types are the basis for thinking about the diversity of hybrid configurations (lines 558-562). But it is something you announce without results and discussions on this hybridization being seen throughout the paper.

What you also mention on intermediate supply chains and the combination of long and short chains (lines 572-578) is interesting. But does your approach allow for this type of configuration to be taken into account? We would like these points to be addressed in the discussion section.

Line 519: typo (scale)

Table 1

It is difficult to know how the barriers were identified. Is it the formulation proposed by individuals, a collective, the innovation team? Why are some barriers very similar in their formulation, see overlapping barriers such as barriers 1, 4 and 7 which focus on the lack of technical knowledge or barriers 22 and 25 on the lack of investment in equipment. We need to give more explanations on this.

It is also surprising to see that CAPs, environmental and sanitory regulations have been grouped together in the same barrier because these regulations refer to very different fields of application and impacts.

Figure 2: see above the remarks on the methodology section and adapt the figure accordingly

Figure 3: the boundaries of the groups are strongly impacted by the barriers mentioned a few times, especially for building outside.

Another question that can be asked, in relation to the paper issue: what information can be extracted from barriers that are not within the limits of a group, such as process_invest_1? is it a barrier that is intermediate between 2 ideo-types?

6. PLOS authors have the option to publish the peer review history of their article (what does this mean?). If published, this will include your full peer review and any attached files.

Reviewer #1: No

Reviewer #2: Yes: Bálint Balázs,Environmental Social Science Research Group, Budapest, Hungary - balazs.balint@essrg.hu

Reviewer #3: No

---

## [Author Response · Author response to Decision Letter 0]

19 Nov 2019

Dear academic editor and reviewers,

We thank you very much for your very constructive and relevant feedbacks on our manuscript. We have done our best to take them into account. We have added a sentence in the acknowledgement part at the end to thank you for your contribution:

“We thank the academic editor and the three reviewers for their complementary constructive feedbacks on our paper. “

Please find below explanations of the changes me made based on your remarks (in italic).

ACADEMIC EDITOR: Your manuscript has good potential to contribute to the scientific community, but it needs substantial improvement to be accepted by PLOS ONE. Despite differences in the overall evaluation of your manuscript between the three reviewers, all of them ask for a clearer and more explicit description of the research design and methodological approach. Furthermore the structure of the manuscript and specific sections need to be improved, as recommended by reviewer 1 & 3. Finally, you need to respond to the fundamental criticism of reviewer 3 doubting the issue of hybridization between niches and dominant regime being approached thoroughly in your study. Also the "imbalance" in your available case studies, with the large majority dealing with one of 27 possible combinations in your conceptual approach needs to be thoroughly discussed.

We will address the points raised by the academic editor (structure, methods, and fundamental criticism) directly after the corresponding feedbacks of reviewers.

Our internal editors have looked over your manuscript and determined that it is within the scope of our Future Crops Call for Papers. This collection of papers is headed by a team of Guest Editors for PLOS ONE. The Collection will encompass a diverse range of research articles on enhanced agronomic production, guaranteeing food security and neglected crop species. Additional information can be found on our announcement page: https://collections.plos.org/s/future-crops.

If you would like your manuscript to be considered for this collection, please let us know in your cover letter and we will ensure that your paper is treated as if you were responding to this call. If you would prefer to remove your manuscript from collection consideration, please specify this in the cover letter.

We would like very much to contribute to the “Future Crops” collection so will indicate it in the cover letter. 

3. We note that Figure 2 includes an image of a patient / participant in the study.

We have informed the individual of the picture and got his signed consent that we keep in our research notes. In part 2.2, we have integrated in the text the following sentence: “The individual appearing here has given written informed consent (as outlined in PLOS consent form) to publish these case details.”

5. Review Comments to the Author

#REVIEWER 1

The authors question the paradigm and dichotomy of aiming at innovations either within the dominant farming systems or outside those. The authors use crop diversification as a case for agroecological innovation and sample barriers towards adopting different forms of it. As the study is not only about innovation systems, but also explicitly about where these are embedded in, the authors introduce the more comprehensive concept of “food system innovation settings”. In this way, they draw attention to the manifold frame conditions of innovations and to the complexity inherent to each innovation system as such. By doing so, the authors build evidence for the targeting of policies aimed at effectively supporting change. In particular, the comparative perspective on different farming styles (conventional, organic) in a sufficiently detailed qualitative analysis makes the manuscript an enjoyable piece of work.

Overall, it is an interesting and solid piece of work. The authors have carefully handled their results. The methods chapter needs some improvement as does the discussion. In general, the manuscript is quite readable. The authors require to check some format and language issues.

Thank you for this feedback, we will address below the specific points you raised.

More details:

One part of the methods needs more information. The case selection apparently simply follows a project rationale. This is per se acceptable but some explanation why the project is/was working on such cases would be needed – and coming to this point just in the discussion is far too late. 

We have added that sentence in 2.1: “In the rationale of this European project, the 25 cases were initially selected to cover a wide range of situations as far pedoclimatic conditions and diversification farming strategies were concerned. This initial selection did not account for types of value chains and/or agriculture (organic or conventional), which explains that the cases design is not optimal for the variables considered in this study.”

What is even more required is some more insight into the type of data collected. Was it mere observation during the course of the project? or was there any maybe structured assessment of information from each “innovation team”? Suddenly, you mention “interview content” (L202) – what you did with whom needs to be clearly outlined. Figure 2 is helpful, though not enough as a stand-alone. Even the discussion (far too late) suddenly reveals some more details about the methods (L488-9). 

We have given more details about the data collection in part 2.2:

“In the initial phase of the project, 5 workshops were organised involving each time 5 different innovation teams. Each team was involved individually in a 2-hours brainstorming based on the drawing of “problem trees” (44, 45). This method aimed at investigating the different barriers that could limit or imped the diversification process, looking each time for the causes behind the actual difficulties faced in each context. During the brainstorming exercise we took notes of the discussions within each innovation team and of the information drawn on the “problem tree”. A synthetic report was also produced by each team to provide an overview of issues identified. It has to be noticed that innovation teams’ leaders and monitors mostly belonged to farming related organisations (farmer’s associations, agricultural R&D, extension or advisory services), which may create a bias in the perception of barriers at different levels of the value chain (see discussion). In a second round (4 months later in average), a short 30-minutes complementary interview was carried out with every innovation team. Based on the first report they produced, participants were asked to deepen the description of barriers to diversification they faced in their specific context. Interviewers ensured that all levels of value chains were covered: (i) farmers and production, (ii) downstream operations from farm to retailing, (iii) marketing and consumers, (iv) contracts and coordination between actors. Those 4 categories, with a specific focus on coordination, where inspired by a previous study on factors impeding crop diversification (16). We aggregated the notes taken during complementary interviews and brainstorming with the information contained in the report produced by each team to build the final “qualitative material” of our study. We carried out qualitative analysis of this material using thematic coding and matrix tools (46, 47). We built a matrix with the presence/absence of 46 barriers (Table 2) across the 25 innovation settings.” 

Comprehensive information about data collection is most relevant when reading the results chapter, where at the beginning of that section it is not always clear whether you are reporting results or are already discussing those. However, this remark does not apply to the sections 3.2.1-3, where the results are nicely referenced and clear.

We acknowledge that in the initial version of this manuscript the first paragraphs of results (3.1 and introduction of part 3.2) were confused, mixing results and discussions. We therefore deleted there the discussion points. The confusion also came from the fact that the initial title of the part 3.2 was not well positioned. We put that title at the right place in the text. 

There are few incidences with choice of terms. For instance, L240 “factors are correlated” – the choice of the word correlated suggests that it is a matter of statistics, what it is apparently not at this point. More confusingly, this incidence finally is from a reference, while it is placed in the beginning of the results section.

We have deleted this sentence which did not add any information and was more a discussion point. 

While the informative sub-titles guide well through the results, the sequence of presenting the results is not always clear to me. For instance, in 3.2. you start with the markets – why about markets first? 

This point was due to the fact that we made a mistake in the place we originally put the title 3.2… The results about markets are at the end of 3.1 and not at the beginning of 3.1. We changed that. 

(L259: not Tab 1 but Tab 2 meant?) 

Well spotted. It is indeed Tab 2; We changed that.

Another example, in Table 2, you already introduce the three “ideal types”, but how you arrived at those comes later. I think you should have presented the MCA earlier. 

We have added a short introduction at the beginning of section 3 (results) to present briefly the ideal-types before Table 2 is mentioned: 

“In part 3.1, we will provide a first general description of barriers to crop diversification highlighted in this study. The 3 ideal-types of food system innovation settings: (“Changing from within”, “Building outside”, “Playing horizontal”) and the barriers they are specifically linked to will be detailed in part 3.2. “

L266-268: this is discussion, not results. 

You are right we deleted those 2 lines (this point was already mentioned in the discussion part anyway).

The first section of 3.2 (before starting with 3.2.1) needs thorough revision, or complete rephrasing. I would introduce the MCA result here instead of just mentioning them casually in L278. 

We think that having put the 3.2 title at the right place and taken off the discussion part make the beginning of 3.2 clearer. As you suggest, we introduce now the MCA results (Fig 3) there:

“Nevertheless, we were able to link 37 barriers more specifically to 3 ideal-types of food system innovation setting (Fig 4).”

The Power barrier is quite discriminating in axis 1, Quant in axis 2, etc., a lot of zero-labeled codes in the middle.

We hear that this may be a suggestion to give an overall description of MCA figures. We think that those points can be seen directly looking at the Figure and do not bring useful information to the reader as far as ideal-types are concerned. So we have not introduced more detailed description of the MCA figure not to make the text too “heavy”. And also because we describe already in details the presence and absence of barriers (with 0 and 1) in part 3.2 1-3. 

What finally remains open about the ideal types is why you decided to have three ideal types and not four, or any other number. The remainder of the non-associated data points also seems to cluster together. How did you form the ideal types? There is some information given in Fig 3, but the text is incomplete on that issue. It is good to avoid repetition in a manuscript, but the text needs to be comprehensible also without the notes in the tables and figures.

We have now provided more insights on those aspects in the text, in the methodological section (2.3):

“Based on the MCA outputs, we visually connected barriers to three “ideal-types” of food system innovation settings (described in section 3) which were specific combinations of innovation modalities described by the supplementary variables. The number, visual determination and limits of those groups was guided by the qualitative analysis of material collected with the different innovation teams.”

Moreover, it is remarkable that the four first axes of the MCA only yield about 40% of the total variance. Similarly, the formed clusters are not obvious regarding the layout of the data points. This requires some critical discussion.

We have added discussions on those points in the section 4.1:

“The visual “drawing” of ideal-types boundaries and connected barriers on MCA outputs (Fig 4) was subject to our human interpretation to “make sense” of data (informed by the qualitative analysis of material collected with innovation teams). This exploratory approach could be enhanced by more systematic methods of clustering on a larger dataset. The impact of less frequent barriers on ideal-types could also be tested while comparing MCA outputs integrating only more frequent barriers. In our analysis, the first four dimensions (axis 1 to 4) of the MCA only explain 42% of total variance. This suggests that although innovation settings play a role in shaping the barriers to crop diversification, other factors should be considered.”

L482-5 makes an outlook to methodological development. I think it is more than that, it is about the usefulness and applicability of the results, the next steps when aiming to transform the scientific results into planning action, either in a scientific or policy attempt. It should be further discussed in this sense. L540ff appears to be somewhat connected to the same issue. You shortly mention recommendations and action priorities that are shown in (the hardly readable) Fig 4.

We have moved this discussion part to the last discussion section (4.3) and elaborated on it. 

“In this study, we characterised each barrier in a binary way (present or absent). To inform policy planning and innovation strategies, a deeper understanding of barriers seems required, especially of their relative “limiting power” (totally blocking, partially limiting, etc.), and of the possibility/necessity to remove it or to adapt innovations if they are not likely to be solved.”

We agree that Fig 5 (replacing Fig 4 because we added a new figure 3) is hardly readable on the pdf.. but the initial figure we sent is well readable. The problem comes from the transformation into pdf made automatically by the online platform. Hopefull, if the paper gets accepted, the figure will appear in higher quality. 

The authors mention variables that could have further sustained their analysis, such as farm size. One factor that is already recorded is the country. Surprisingly, the authors mention some differences but do not do so in a systematic manner. It remains unclear why that factor has not been further followed up.

We chose not to integrate the size of farms and the country (11 possibilities) in our analysis because the possible modalities for such variables were not well balanced across the sample. For example, there was only one case in Germany and 4 in France. We thought that such imbalance would not lead to a robust analysis of the impact of such factors. Moreover, other studies in our project (based on more cases specifically focused on those factors). In this study, we really wanted to focus on farming practices and value chain orientation. 

We have added a sentence in the discussion part to mention that. 

The conclusions read very well. There is only one point that needs attention: power asymmetries and games. They come a bit out of the blue and are then prominent in your final statement. Please better connect to your data, maybe even need to present additional data in the results section.

We have totally reformulated our conclusion (also taking into account feedbacks of reviewer 3). The discussion on hybrid food systems appear now in a new discussion section (4.4) and the conclusion is more focused on results and more reasonable results-ground perspective. 

Overall, the manuscript is quite readable. Nevertheless, there are still some linguistic challenges regarding the employ of terms and punctuation, and to some extent grammar and spaces.

 Examples: Terms, L108 the crops “are very contrasted”,

We changed into “much contrasted”.

 L71 Rotations are “enhanced”, 

We changed into “improving crop rotations”.

L129 “screening” of harvested crops. 

We use now the term “separation” of harvested crops and have modified all occurrences of “screening” in the text, figures and tables. 

Punctuation, L70 to promote “omega 3 rich products”: many words in a row that could be better structured through hyphens, 

We changed into omega-3-rich products.

also L161, also outside niche innovations an on plot grazing.

The reference to “outside-niche” innovation was deleted because that sentence was deleted from the conclusion to integrate reviewers’ feedbacks (keeping conclusion closer to results). We changed on plot grazing to “on-plot” grazing.

Grammar: L183 multi-step procedure (not steps here). These are examples, the incidences are not limited to these. 

We changed into “multi-step” procedure as suggested and modified other incidences of similar “multi-word patterns”

Moreover, the reference list still contains some French and there are several issues with upper/lower case writing. 

We corrected issues of upper/lower case in the list of references. There are only 2 references left in French (15 and 53) because we were not able to find corresponding ones in English. 

Table 1 is an example of raw data that needs some processing by the authors: some sorting is required – what is the rationale behind the current sequence of rows? 

We modified Table 1 and ordered now the rows by “Diversification strategy”, “Value chain” and “Type of agriculture”. We have added in the legend that the “Case number” allows to find more information and description about the corresponding case on the project website. 

In the text, the use of the normal brackets instead of the square ones for indicating reference is somewhat irritating – and even more, when other numbers are also put in such brackets (L230). 

We now use square brackets to indicate references in the text.

There are some very long sentences (up to 5 lines), which need structuring through breaking down to smaller sentences.

We checked in the text and made shorter sentences when required.

 L397-8 lacks a verb – incomplete sentence. 

The initial sentence did have a verb: “Strategies of (i) spatial crop diversification and (ii) arrangements between farmers to support new crops face similar barriers, balanced between farm level and other levels of value chains (Fig 3, Table 2).”

However, to make it easier, we changed the sentence into:

“Similar barriers, balanced between farm level and other levels of value chains, were faced by strategies of (i) spatial crop diversification and (ii) arrangements between farmers to support new crops, (Fig 4, Table 2).”

Small but intriguing: L585 not processes but processors meant? 

We meant “Innovative food processing technics” and changed it in the text.

MLP does not deserve to be abbreviated, it is too much spread over the manuscript and not often used. 

Ok, we changed it. 

From what I learned, English sentences never start with a figure – always spell out numbers at the beginning of a sentence.

Ok, we checked in the text and changed it when required.

 The font size of the figures, in particular number 4, is quite small and probably too small. The figure 4, as reproduced for the review, is hardly readable; what is the meaning of the different arrow types and the different sizes of the elements? 

The initial figures we provided had higher quality resolution and were readable. This is the transformation into a pdf for the review which made them hardly readable. We hope that it will be better in the editing process if our paper is accepted (especially if figures are presented in landscape format as intended). If required at that stage, we could provide figures with larger font. 

In Fig 5 (new Fig 4), we added the following details in the legend:

“Arrows represent the flows of products from production to consumption along the value chain. The size of value chain actors is in line with the average size of actors in the different ideal-types, which tended to be smaller in local and organic value chains.” 

Table 2: is the color in the right columns the same as a “1” in the codes? If so, then better use “1” in both cases.

Yes, the colour corresponds to “1”. We have therefore integrated the “1” codes in Tab 1 to unify the representation but left the colours corresponding to the value chain levels because we find that it helps the reader to get a quicker grasp on how barriers are spread along value chains in each case. We have made this choice clearer in the legend of Table 1. 

 In general, the authors have made considerable efforts to provide comprehensive legends, which is highly appreciated.

#REVIEWER 2

The authors aim at presenting an evidence base for analysing innovations in the food systems fostering crop diversification. The central claim is that by identifying patterns of barriers of innovation along the value chain in various settings, the authors will point towards better-targeted research and policy actions. The main strength of the paper is that it goes beyond the dichotomist perspective of the food system and develops a typology of food system innovation, taking into account the hybridisation of food systems.

Thanks for this overall feedback.

I would suggest the following changes for more clarity for non-specialists:

- start with one single research question and tell why this is interesting to study - how innovation can start and what is its locus (farm, chain or system)?

To gain clarity for non-specialists, we re-organised and added details to the introduction. After defining better the food systems and highlighting that innovations can happen at every level, we present now a clear question before further elaborating on the theoretical questions and concepts around innovations, and the different type of innovation settings.

That clearly formulated question is:

“To which extent are the barriers to sustainable innovation at different levels of food systems dependant on innovation strategies and contexts?”

- add more details on your theoretical standpoint and the research design

To clarify our theoretical standpoint, we added this sentence to the introduction:

“Our analytical framework combines (i) concepts from the multi-level perspective on socio-technical transitions (13–19) to investigate innovation settings according to their proximity with the dominant regime and (ii) recent agroecology literature arguing that supporting innovation in food systems should rely on a preliminary characterization of their different interrelated components (2).”

Details on how we mobilise those different elements are given in part 2. 

In the introduction, we added details on the research design and more generally on the methods we used in the section 2.1:

“In the rationale of this European project, the 25 cases were initially selected to cover a wide range of situations as far pedoclimatic conditions and diversification farming strategies were concerned. This initial selection did not account for types of value chains and/or agriculture (organic or conventional), which explains that the cases design is not optimal for the variables considered in this study.”

And in the section 2.2:

“In the initial phase of the project, 5 workshops were organised involving each time 5 different innovation teams. Each team was involved individually in a 2-hours brainstorming based on the drawing of “problem trees” (44, 45). This method aimed at investigating the different barriers that could limit or imped the diversification process, looking each time for the causes behind the actual difficulties faced in each context. During the brainstorming exercise we took notes of the discussions within each innovation team and of the information drawn on the “problem tree”. A synthetic report was also produced by each team to provide an overview of issues identified. It has to be noticed that innovation teams’ leaders and monitors mostly belonged to farming related organisations (farmer’s associations, agricultural R&D, extension or advisory services), which may create a bias in the perception of barriers at different levels of the value chain (see discussion). 

In a second round (4 months later in average), a short 30-minutes complementary interview was carried out with every innovation team. Based on the first report they produced, participants were asked to deepen the description of barriers to diversification they faced in their specific context. Interviewers ensured that all levels of value chains were covered: (i) farmers and production, (ii) downstream operations from farm to retailing, (iii) marketing and consumers, (iv) contracts and coordination between actors. Those 4 categories, with a specific focus on coordination, where inspired by a previous study on factors impeding crop diversification (16). We aggregated the notes taken during complementary interviews and brainstorming with the information contained in the report produced by each team to build the final “qualitative material” of our study. We carried out qualitative analysis of this material using thematic coding and matrix tools (46, 47). The general aim of this approach was to build more and more abstract categories on the basis of an iterative

cross analysis of interview contents. This resulted in categorising 46 barriers (Table 2) to crop diversification. The level of abstraction to characterise barriers was linked to the level of precision used by innovation teams to describe them. For example, innovation teams discussed very specifically many aspects of the lack of knowledge and references for farmers. This is the reason why we chose to distinguish 3 specific categories of related barriers related to this lack of knowledge: about technical implementation of farming practices (K_Tec), impact of new practices on the sustainability of the farm (K_Sustain), impact of new practices on the global design of the farming system (K_Syst). Conversely, for other barriers, innovation teams mentioned very generic challenges with limited precision, which resulted in broader categories. For example, we considered a single category for challenges related to common agricultural policy (CAP), environmental or sanitary regulations although it may embrace various fields of application and impacts. 

We built a matrix with the presence/absence of 46 barriers (Table 2) across the 25 innovation settings. “

- original data should be available in Zenodo or similar repository

The original data are available on the following Zenodo repository as indicated in the text: https://zenodo.org/record/3249967#.XQoQoo_gpPY

- introduce your definition of the food system, food regime - e.g.line 53.

We have now introduced earlier in the introduction a definition of the food system followed by the definition of food regime.

- provide more detail of the case studies and the "barriers" workshops for the curious reader: visuals, verbatims, participatory observation notes or drawings, etc.

In order the reader to be able to get more details about the different cases, we have added the following sentence and link: “Details about the 25 cases can be found at https://www.diverimpacts.net/case-studies.html using the case number indicated in Table 1.” 

As indicated above, we now provide a more detailed description of the research design, workshops and interviews. We now present in the supplementary material (S1 Appendix) a new document with guidelines provided to innovation teams to prepare complementary interviews. 

We added a figure (Fig 3 now) with 2 pictures to illustrate the brainstorming phase in the first round and the complementary interview in the second round. 

We do not wish to provide pictures of the “problem trees” drawn by the participants or notes taken during the workshops because they present sensitive data (name of specific companies or actors). 

- more reflection on self-referentiality: how your role as researchers can change the innovation settings.

We have added this paragraph at the end of part 4.2:

The role that researchers can play in innovation dynamics at the value chain level should also be better explored. During the workshops, we (as scientists) provided guidelines to innovation teams to support them in identifying the different barriers they face (S1 Appendix). This framework was based on preliminary literature review and underlined the necessity to consider multiple dimensions of barriers at all levels of value chain. It also provided a first list of potential barriers to explore with stakeholders. This framework may have helped to broaden innovation teams’ perspective beyond the aspects they would normally consider in their professional activities (mainly dealing with farm level issues). On the other hand, it may have conditioned the way innovation teams interacted with other value chain actors and could have limited the exploration of dimensions not previously mentioned by literature. In this regard, it seems important for scientists to develop reflexivity in exploring acceptable trade-offs between (i) providing scientific information to accelerate/broaden change processes with a risk of pre-conditioning too much innovation pathways and (ii) playing only a role of facilitators that may allow more original development but do not make actors benefit from previous scientific knowledge.

#REVIEWER 3

As it stands, this article requires many improvements in order to be published.

The authors' thesis is that it is necessary to go beyond a dual approach to transition and innovation processes that would distinguish a dominant regime with incremental innovations and niches that support radical innovations. According to them, the literature most often favours an in-depth analysis of one of the two situations without considering the possible hybridizations and interconnections of the two. There is therefore a lack of comparative and large-scale approaches to better understand barriers to innovation in a diversity of contexts that do not strictly speaking rely on a dichotomy between dominant and niche actors. This is therefore what the authors propose to do by addressing the barriers to crop diversification along agri-food systems (production, collection, processing, retail, market and actors coordination). The authors use 25 European case studies as a basis for this. And they introduce the concept of food system innovation settings combining a type of innovative agricultural practice, a type of value chain supporting this innovation and a type of agriculture (organic or not).

My main criticism is that I do not see the added value of their work on the issue of hybridization between the dominant regime and niche actors. The authors point out that this has never been done in the literature, but I am not convinced that they really address the issue of hybridization between a dominant regime and innovative niches. On the theoretical level, however, there is a whole literature that has addressed the question of the articulation between dominant regime and niche, starting with the multi-level perspective, which is also mentioned many times in this article. Also it seems to me that on the one hand the authors minimize the contributions of these studies on hybridization between niches and dominant regime. On the other hand, the authors use a very normative approach, consisting in categorizing the social world they are studying, a categorization that should be discussed and which seems difficult to reconcile with a detailed analysis of the hybridization processes to which the authors aspire. The concept of “food system innovation settings” seems more suited to the variability of the case studies mobilized than to a real contribution as a concept for a better understanding of the barriers to transition to more sustainable systems. The following points therefore need to be clarified:

The basic proposal for the “food system innovation settings” is based on 3 criteria with 3 modalities, which implies 27 possible combinations. Are these combinations all possible? How could some of them be subject to more or less strong barriers? How are some of them more or less similar or more or less compatible? Authors must discuss these questions if they want to use these criteria in the service of a new concept.

Thank you for this really fundamental critic on our work. We agree with you. Our work was an attempt to link barriers to diversification to innovation settings based 25 innovation cases. At the end, we highlight 3 ideal-types and discuss them as far as research and policy priorities are concerned. We think that our work provides interesting new element based on this comparison relying on empirical data. However, in this state, our framework does not allow to explore hybridisation in food systems. 

So, we should not pretend we do that and be more precise about the scope of our study. In the abstract and introduction we replaced the following sentence:

“We discuss perspectives to support change beyond the dichotomist distinction “within versus outside” the regime in order to account for the complexity of coexisting hybrid food systems between alternative niches and dominant regime.”

By:

“We discuss further development of our approach to analyse barriers faced in intermediate and hybrid food system configurations.”

In the introduction, material and methods, results and discussions, we keep on describing our 3 ideal-types and the linked barriers. It is only in a new discussion section at the end (4.4) that we open up the discussion about the need to consider hybrid and intermediate food systems. In that section we also address your comments about the fact that our 25 cases were not balanced as all possible combinations of modalities are concerned. 

In that new discussion section, we present how our approach could be further developed to account for hybrid and intermediate food systems. However, we do not suggest any longer that this would allow to analyse hybridisation as a dynamic process as MLP could do. This “dynamic” analysis is out of our scope which is to analyse a diversity of systems and their related barriers:

“Our conceptual proposal of food system innovation setting is based on 3 criteria with 3 modalities which implies 27 combinations. Of the 27 possible combinations, 16 of the 25 case studies (64% of cases) are in 4 of the 27 possible combinations (14.8% of the possible combinations). 

This study was based on empirical data collected on the 25 cases of the DiverIMPACTS project which were not initially selected to cover all possible combinations of innovation settings. In those cases, there is a very strong link between local chains and organic and between conventional farming and commodity value chains. This can be explained by the fact that organic farmers may be more likely to be involved in local value chains. Organic agriculture was developed as an alternative to dominant agriculture and that promoting shorter supply chains was historically a strong strategy to support the development of that innovation niche [10, 22]. The resulting imbalance in possible combination of modalities in our sample implies that we are not allowed in our analysis to distinguish the relative contribution of “organic” and “local” in the “Building outside” ideal-type and of “conventional” and “commodity” in the “Changing from within outside”. In that sense, those two food systems ideal-types echo the dichotomy of radical versus incremental [1, 18, 23], transforming versus conforming [25, 67], alternative versus dominant [10, 68, 69] found in literature. The third ideal-type that we develop, “Playing horizontal”, tends to show that “things may be more complicated out there” and that even as a heuristic tool, binary categorisation may be limited when trying to grasp food systems complexity.

Moreover, recent studies show that hybrid foods systems are developing, e.g combining for example conventional and local chains [73], or organic and commodity chains [55] as in 3 of our 25 cases (Table 1). Would the development of organic intercropping be eased or complicated by the integration in well-structured dominant value chains rather than local ones? Would new crops in conventional rotations face less or more problems if they target local quality markets instead of export as commodities? To disentangle the specific contribution of agriculture type and value chains in such hybrid settings, and potentially creating more complex ideal-types, our methodological approach could be applied to a wider sample of cases ensuring a balanced cover of all possible combinations. 

Intermediate food value chains are also growing, with structured industrial-like organisation but a strong connection to local farmers and farmers’ groups with constraining traceability, fair share of added value and ecological practices, are growing in industrialised countries. The potentiality of combining benefits of long and short supply chains in such hybrid food systems is starting to be explored under the umbrella concepts of “agriculture of the middle”, “mid-tier supply chains” or “values-based food supply chains” [72, 73]. To analyse the specific barriers to diversification in such hybrid chains, we could integrate new possible modalities in our approach, e.g “intermediate value chain”. Another possible research approach would to discuss with actors involved in hybrid value chains about the similarities and differences in the barriers they face compared to our “pure” ideal-types. Doing so, we may be able to highlight whether (i) hybrid configurations face cumulated challenges of the “Changing from within” and “Building outside” categories, (ii) hybrid configurations result in solving of creating more barriers.”

We have now also deleted the discussion on hybridisation and interaction between food systems from our conclusion. Although we think it is very interesting, our study does not either provide elements to address interactions between contrasted food systems. So all the aspects about sharing/appropriating knowledge, co-optation/dilution issues in the interaction between food systems have been deleted. Our conclusion is now more based on our results and more reasonable, results-grounded perspective (see below the discussion part about the conclusion). 

Each criterion is also worth discussing.

Crop diversification practices: what is the scientific justification for these 3 categories? This should be clarified because there seems to be a mix of agronomic and technical criteria. It is in particular the distinction between spatial and intercropping that involves a purely technical criterion, whether or not to screen the harvest, that poses a problem. When is it decided that a technical criterion justifies the creation of a new category? And who decide this?

Sale of products: the authors propose two rather binary categories: commodity with long chains and export and local with short chains and alternative food systems. The introduction of a third category refers to something quite specific related to exchanges between farmers and stockbreeders.

Type of agriculture: the authors distinguish three cases: AB, conventional and both. But why do they distinguish these 3 cases? What hypotheses behind them?

By discussing these questions, it may be easier to understand the value of this grid and how it can be mobilized.

We have added a general paragraph in that section (2.1) to explain how those different categories were built.

“Among the 25 cases, we characterised three categories of diversification practices, value chain and agriculture type based on a preliminary qualitative analysis of interviews carried out with innovation teams (see part 2.2). Those categories were tailored to echo the preliminary links that we were able to observe between the diversity of innovation settings and barriers (before running the multiple correspondence analysis described in part 2.3).” 

For crop diversification practices, we have added information to clarify our choice of categories:

« We chose to distinguish intercropping from other types of crop spatial diversification based on the need to separate crops after harvest because both previous literature [35, 41] and qualitative analysis of interviews show that crop separation bring specific barriers in the cropping design, management and post-harvest phases.”

We did the same for type of agriculture:

“”This categorisation relies on the assumption that conventional farmers may face more barriers to crop diversification at the farm level because the development of conventional agriculture since World War II relied on specialisation. Conversely, crop diversification is a historical pillar of organic agriculture.”

The 25 case studies in the grid: of the 27 possible combinations, 16 of the 25 case studies (64% of cases) are in 4 of the 27 possible combinations (14.8% of the possible combinations). In particular, there is a very strong link between local and organic and between including conv and commodities. A slightly more in-depth analysis is needed to determine why these combinations are more common.

We address now this issue in a new discussion section (4.4) already presented above. 

Methodology section

We lack information on the methodology for collecting case study data. In particular, some elements are introduced into the discussion as limitations when they should be presented in the methodological part. In particular, it is necessary to specify:

- How the case studies were selected?

- How the innovation team was set up?

- Who participated in the 2-hour brainstorming session?

- How was the list of barriers drawn up, given that some formulations are very similar to each other?

As already presented in the answer to previous reviewers’ feedback, we have now substantially developed the description of our methodology (implementation of interviews and workshops, role of innovation teams, formulation of barriers etc.) in part 2.2, addressing issues mentioned by reviewer 3:

“In a first round, 5 workshops were organised involving each time 5 different innovation teams. Each team was involved individually in a 2-hours brainstorming (Fig 3, a) based on the drawing of “problem trees” [44, 45]. This method aimed at investigating the different barriers that could limit or imped the diversification process, looking each time for the causes behind the actual difficulties faced in each context. During the brainstorming exercise we took notes of the discussions within each innovation team and of the information drawn on the “problem tree”. A synthetic report was also produced by each team to provide an overview of issues identified. 

In a second round of workshops (4 months later in average), a short 30-minutes complementary interview (Fig 3,b) was carried out with every innovation team. It has to be noticed that innovation teams’ leaders and monitors mostly belonged to farming related organisations (farmer’s associations, agricultural R&D, extension or advisory services), which may create a bias in the perception of barriers at different levels of the value chain (see discussion). To mitigate that bias, innovation teams were encouraged to contact before complementary interviews as many value chain actors as possible (farmers, processors, retailers etc.) in order to collect information about barriers at different levels of the food system. To support innovation teams in that task, we provided them guidelines and a structured framework of data collection (S1 Appendix). 

During second-round complementary interviews with innovation teams, participants were asked to deepen the description of barriers to diversification at different levels of value chains. Interviewers ensured that all levels were covered: (i) farmers and production, (ii) downstream operations from farm to retailing, (iii) marketing and consumers, (iv) contracts and coordination between actors. Those 4 categories, with a specific focus on coordination, where inspired by a previous study on factors impeding crop diversification [16]. We aggregated the notes taken during first-round brainstorming and second-round complementary interviews as well as the first-round reports to build the final “qualitative material” of our study. We carried out qualitative analysis of this material using thematic coding and matrix tools [46, 47]. The general aim of this approach was to build more and more abstract categories on the basis of an iterative

cross analysis of interview contents. This resulted in categorising 46 barriers (Table 2) to crop diversification. The level of abstraction to characterise barriers was linked to the level of precision used by innovation teams to describe them. For example, innovation teams discussed very specifically many aspects of the lack of knowledge and references for farmers. This is the reason why we chose to distinguish 3 specific categories of related barriers related to this lack of knowledge: about technical implementation of farming practices (K_Tec), impact of new practices on the sustainability of the farm (K_Sustain), impact of new practices on the global design of the farming system (K_Syst). Conversely, for other barriers, innovation teams mentioned very generic challenges with limited precision, which resulted in broader categories. For example, we considered a single category for challenges related to common agricultural policy (CAP), environmental or sanitary regulations although it may embrace various fields of application and impacts. 

 We built a matrix with the presence/absence of 46 barriers (Table 2) across the 25 innovation settings.” 

About the selection of cases, we have added information in part 2.1. 

“In the rationale of this European project, the 25 cases were initially selected to cover a wide range of situations as far pedoclimatic conditions and diversification farming strategies were concerned. This initial selection did not account for types of value chains and/or agriculture (organic or conventional), which explains that the cases design is not optimal for the variables considered in this study.”

Finally, the authors present in a discussion section in 4.2 the fact that barriers result from farmers' perceptions and related advisory services. This is indeed a limitation but it must be presented beforehand in the methodology section.

We have added information on this aspect in the new version of section 2.2 as presented above:

“It has to be noticed that innovation teams’ leaders and monitors mostly belonged to farming related organisations (farmer’s associations, agricultural R&D, extension or advisory services), which may create a bias in the perception of barriers at different levels of the value chain (see discussion).”

Result section

This section is poorly structured. If the 3 ideo-types constitute the main result, they should be highlighted from the beginning of this section.

We have now better structured this result section.

First of all, we have added an introduction paragraph at the beginning. We still prefer to fist present the 46 barriers before describing more specifically the 3 ideal-types. 

“In part 3.1, we will provide a first general description of barriers to crop diversification highlighted in this study. The 3 ideal-types of food system innovation settings: (“Changing from within”, “Building outside”, “Playing horizontal”) and the barriers they are specifically linked to will be detailed in part 3.2. “

General information is introduced before Table 2, including the paragraph just above (lines 242 to 247). To say that logistics is not adapted to small volumes and that new crops are in competition on the global market seems a little dated and not directly related to the results at this stage.

We have now moved the general description of Table 2 after introducing that table. To make clear that those paragraphs are not just general information but a synthetic description of the barriers found, we now precise:

 “Among the 25 cases, major barriers to crop diversification at the farm level are related to the lack of technical knowledge and references regarding minor crops and crop diversification….. “

What we say about the competition on the global market and on logistics come from the analysis of the interviews and were generally mentioned by many innovation teams (Compet: 9 cases; for example Coll_Vol: 16 cases). As here we present a synthetic view of main barriers among the 25 cases presented in Table 2, we think it is ok to mention it here. 

The first 2 paragraphs of 3.2 seem out of scope. The first (lines 258-265) is focused on market levers and the second (lines 266-268) is a reminder of the merit of this work.

The presentation of the 3 ideo-types is clear (lines 274-457).

The initial 2 first paragraphs of 3.2 seemed out of scope because we made a mistake in the positioning of the title of section 3.2. Those 2 paragraphs are in fact the end of section 3.1 (describing the last type of barriers in Table 2). We fixed that.

Discussion section

The discussion seemed more a reminder of the limitations of this work than a real discussion. I think it is possible to go further by questioning these 3 ideo-types in relation to the grid proposed at the beginning.

We discuss now the possible use and further development of our approach compared to the possibilities of the grid proposed at the beginning in a new discussions section 4.4 (as already explained above). 

Section 4.1: The main message you are sending is that your results should be considered as not very robust (lines 466-471) since some barriers only concern a small number of cases. Is it possible to go a little further? In particular, it seems that it is the Building outside strategy that is most affected by barriers based on a small number of cases

We now discuss the way the drew the boundaries of ideal-types and suggest perspectives to improve the method in section 4.1. 

“The visual “drawing” of ideal-types boundaries and connected barriers on MCA outputs (Fig 4) was subject to our human interpretation to “make sense” of data (informed by the qualitative analysis of material collected with innovation teams). This exploratory approach could be enhanced by more systematic methods of clustering on a larger dataset. The impact of less frequent barriers on ideal-types could also be tested while comparing MCA outputs integrating only more frequent barriers. “

Section 4.2: Yes, you need to recall the limitations of barrier identification and how to improve barrier identification. 

We have added a new paragraph recalling for the new elements about barrier identification we mentioned in part 2.2 and propose possible improvements. 

“We already mentioned in the methodological section (part 2.2) that the precision in the description of barriers was impacted by the fact that innovation teams were farm-oriented. To ensure a similar level of precision in the characterisation of barriers at all value chain levels, which may for example result in creating more detailed categories for barriers after the farm gate, we could carry out complementary interviews with other value chain stakeholders, which may have more expertise on those aspects.”

We now also discuss possible improvements of barrier characterisation in section 4.3

“In this study, we characterised each barrier in a binary way (present or absent). To inform policy planning and innovation strategies, a deeper understanding of barriers seems required, especially of their relative “limiting power” (totally blocking, partially limiting, etc.), and of the possibility/necessity to remove it or to adapt innovations if they are not likely to be solved.”

Your analysis of the agency level seemed a little caricatural to me. Also avoid referring to article under review (60).

We agree that our analysis of the level of agency is a little caricatural and would deserve finer analysis. This is the reason why, we now say:

“it MAY inform us about the level of agency” or “it MAY denote…”

The article of reference 60 has been published now, so we have updated the reference list. 

Section 4.3: you recall that the literature has focused on a type of dominant food system versus alternative food system situation, most often at national levels, but finally what is the added value of your approach compared to existing ones?

We have reformulated this paragraph and added a sentence to highlight that the added value of our work relies in the comparative approach of contrasted food systems rather than focusing on one type of innovation strategy as previous work did:

“The main added value of our work is to provide a comparative analysis of barriers faced by different food system innovation dynamics that have previously been investigated in distinct studies either on (i) lock-in situations preventing sustainability innovations in dominant food systems [12–16, 31, 35–38, 66], (ii) impeding factors and success conditions for alternative innovation dynamics (born outside the dominant system) to develop [1, 9, 22, 24, 26–28], or (iii) challenges to support crop-livestock integration at the territorial scale, especially for the “Playing horizontal” ideal-type [42, 43, 56].”

We insist now on the fact that this is that comparative dimension allow to formulate specific recommendations:

“The comparative dimension of our work across a diversity of food systems allows to show that different innovation settings in terms of agricultural practices and value chain can result in contrasted patterns in the combination of barriers. This diversity of challenges has to be taken into consideration to develop targeted research, innovation and policy actions according to the food systems they seek to support. “

Lines 525-535: What you are saying has already been said in the literature

We have deleted those lines, as you are right, they did not bring any new element. 

Section conclusion

It is necessary to recall your main results

Why so many questions in the first paragraph (lines 562-567)

There are a lot of references in the conclusion, see in particular the last paragraph.

You argue that your ideo-types are the basis for thinking about the diversity of hybrid configurations (lines 558-562). But it is something you announce without results and discussions on this hybridization being seen throughout the paper.What you also mention on intermediate supply chains and the combination of long and short chains (lines 572-578) is interesting. But does your approach allow for this type of configuration to be taken into account? We would like these points to be addressed in the discussion section.

We have totally redesigned our conclusion with a recall of our main results and not formulating many more questions and not mentioning new references. 

“Based on the analysis of 25 European cases promoting crop diversification, we propose to characterise food system innovation settings as combinations of farming practices, agriculture type and value chain. In this study, we highlight three ideal-types of food system innovation settings: (i) “Changing from within” where longer rotations are fostered on conventional farms involved in commodity supply chains, (ii) “Building outside” where crop diversification integrate intercropping on organic farms involved in local supply chains, (iii) “Playing horizontal” where actors promote alternative crop diversification strategies, either strictly speaking horizontal at spatial level (e.g. strip cropping) or socially horizontal (arrangement between farmers) without challenging directly the vertical organisation of dominant value chains.

Each ideal-type is linked to a specific pattern of barriers to crop diversification. For example, in the “Changing from within” ideal-type, farmers aim to develop innovations that can be compatible with existing infrastructures and norms of big agro-industry players, which explain that the highest number of barriers are related to the production level, such as developing knowledge and management tools to integrate new crops in historically simplified and short-term profit-oriented production systems. Conversely, the “Building outside” innovation setting does not seek to fit in the dominant regime. The majority of barriers in that case are thus rather related to the building of new value chains and to operations after production such as developing adapted technology for the post-harvest management and processing of new crops at small scale. In the “Playing horizontal” idea-type, post-harvest operations are not mentioned as limiting factors because the selling and processing of crops either aim to fit the dominant regime or are managed at farm scale. Barriers are rather related to changes required in cognitive, regulatory and administrative frames to ease spatial innovations at new scales (crop strips on the farm, territorial collaboration between farmers). 

This work allows a better understanding of the specific barriers that should be considered to develop targeted research, innovation and policy actions according to the food systems they seek to endorse. It contributes to recent research developments aspiring to better analyse the diversity of food systems to support transition [2, 39].”

The questions initially formulated in the conclusion about hybrid and intermediate food systems are now addressed in the discussion part with a new section (4.4) where we go more deeply in exploring to which extent our approach could be further developed in that sense. 

Line 519: typo (scale)

We have added the missing “e” to “scale”.

Table 1

It is difficult to know how the barriers were identified. Is it the formulation proposed by individuals, a collective, the innovation team? Why are some barriers very similar in their formulation, see overlapping barriers such as barriers 1, 4 and 7 which focus on the lack of technical knowledge or barriers 22 and 25 on the lack of investment in equipment. We need to give more explanations on this.

It is also surprising to see that CAPs, environmental and sanitory regulations have been grouped together in the same barrier because these regulations refer to very different fields of application and impacts.

We have provided more information on that in the section 2.2

“We carried out qualitative analysis of this material using thematic coding and matrix tools [46, 47]. The general aim of this approach was to build more and more abstract categories on the basis of an iterative

cross analysis of interview contents. This resulted in categorising 46 barriers (Table 2) to crop diversification. The level of abstraction to characterise barriers was linked to the level of precision used by innovation teams to describe them. For example, innovation teams discussed very specifically many aspects of the lack of knowledge and references for farmers. This is the reason why we chose to distinguish 3 specific categories of related barriers related to this lack of knowledge: about technical implementation of farming practices (K_Tec), impact of new practices on the sustainability of the farm (K_Sustain), impact of new practices on the global design of the farming system (K_Syst). Conversely, for other barriers, innovation teams mentioned very generic challenges with limited precision, which resulted in broader categories. For example, we considered a single category for challenges related to common agricultural policy (CAP), environmental or sanitary regulations although it may embrace various fields of application and impacts. “

Figure 2: see above the remarks on the methodology section and adapt the figure accordingly

As already mentioned, we have now developed a more detailed paragraph about our methodology in section 2.2. The Figure 2 only presents a synthetic view of the process. We have added a new Figure (Fig 3) to provide more visual insights on the implementation of workshops. 

Figure 3: the boundaries of the groups are strongly impacted by the barriers mentioned a few times, especially for building outside.

Another question that can be asked, in relation to the paper issue: what information can be extracted from barriers that are not within the limits of a group, such as process_invest_1? is it a barrier that is intermediate between 2 ideo-types?

We now discuss the way the drew the boundaries of ideal-types and suggest perspectives to improve the method in section 4.1. 

“The visual “drawing” of ideal-types boundaries and connected barriers on MCA outputs (Fig 4) was subject to our human interpretation to “make sense” of data (informed by the qualitative analysis of material collected with innovation teams). This exploratory approach could be enhanced by more systematic methods of clustering on a larger dataset. The impact of less frequent barriers on ideal-types could also be tested while comparing MCA outputs integrating only more frequent barriers. “

About the barriers that are not within the boundaries of ideal-types, we mention in section 3.2:

“Some barriers are not specifically linked to innovation settings, for example the lack of technical knowledge and references for implementing new crops that can be considered as transversal across cases (Table 2, Fig 4).”

And in the section 4.3, we have integrated a new discussion paragraph:

“Those specific R&D and policy actions could be articulated with ones targeting barriers that are not linked to one innovation setting in particular and can then be considered as more generic. This is the case for barriers outside of ideal-types boundaries (Fig 4), shared across two innovation setting or not specifically linked to any of them (Table 2). For example, such transversal interventions could foster the development of technical references for growing new crops (K_Tec) and the access to seeds and varieties of minor crops adapted to a diversity of local conditions (Varieties, Seeds). They could also support investment in post-harvest and processing facilities (Pre_ProInvest, Process_Invest) and encourage suitable contracts between value chain actors to share the risks associated with the variability of production, especially during the first years when farmers are experimenting new practices (Variab).”

---

## [Decision Letter · Decision Letter 1]

3 Jan 2020

PONE-D-19-17348R1

Innovation within or outside dominant food systems? Different challenges for contrasted strategies of crop diversification in Europe

PLOS ONE

Dear Dr Morel,

Thank you for submitting your revised manuscript to PLOS ONE and considering all relevant comments raised by the reviewers of your manuscript. After careful consideration, we feel that it has merit but does not fully meet PLOS ONE’s publication criteria as it currently stands. Therefore, we invite you to submit a revised version of the manuscript that addresses the points raised during the review process.

Please consider all relevant language issues raised by the two reviewers. It is recommended to consult a professional native speaker for general language editing to further improve the clarity of your manuscript.

We would appreciate receiving your revised manuscript by Feb 17 2020 11:59PM. To enhance the reproducibility of your results, we recommend that if applicable you deposit your laboratory protocols in protocols.io, where a protocol can be assigned its own identifier (DOI) such that it can be cited independently in the future. For instructions see: http://journals.plos.org/plosone/s/submission-guidelines#loc-laboratory-protocols

We look forward to receiving your revised manuscript.

Kind regards,

Til Feike, PhD

Academic Editor

PLOS ONE

Reviewers' comments:

Reviewer's Responses to Questions

**Comments to the Author**

1. If the authors have adequately addressed your comments raised in a previous round of review and you feel that this manuscript is now acceptable for publication, you may indicate that here to bypass the “Comments to the Author” section, enter your conflict of interest statement in the “Confidential to Editor” section, and submit your "Accept" recommendation.

Reviewer #1: All comments have been addressed

Reviewer #3: All comments have been addressed

2. Is the manuscript technically sound, and do the data support the conclusions?

Reviewer #1: Yes

Reviewer #3: Yes

3. Has the statistical analysis been performed appropriately and rigorously? 

Reviewer #1: Yes

Reviewer #3: Yes

4. Have the authors made all data underlying the findings in their manuscript fully available?

Reviewer #1: Yes

Reviewer #3: Yes

5. Is the manuscript presented in an intelligible fashion and written in standard English?

Reviewer #1: No

Reviewer #3: Yes

6. Review Comments to the Author

Reviewer #1: The authors have well responded to the comments. Still, a thorough language check is needed. The text is comprehensible but there are still so many incidences where the text is not clear enough, the used terms probably not always the correct ones or where grammar issues cause some confusion (e.g. prepositions, the use of plural/singular in composed terms, commas, using “the”, and the s in the present tense). In the following some detailed comments – but I stopped with marking language issues after some lines.

L26-28 instead of always writing “a” type of, use “the” type of

L32 where crop diversification increases (with s)

L43: The transition … requires (with s)

L73: of farmers taking the related risks (not whatsoever risks)

L82: approaches (with s)

L89-92: complicated sentence, check with a native

L101-103: be more specific, check the correct wording. Why is “reduced inputs” a sustainability benefit? Did you mean something like relying less on external inputs? “yields gaps” – closing yield gaps? I think the s from risks is not needed here. Normally we use the term ecosystem services – did you mean this by ecosystemic services, or is that something else?

L103: have proven

L104: of “the” food systems

….am more or less stopping here with my specific language, grammar etc notes…

L119: lens? Is that lentils?

Tab 1 (and Tab2): in the first row of Tab 1, better use Nb as used in Tab 2, good for consistency, instead of number. Even though the legend explains it, number may cause people think that you talk about the number of cases, such as 3 cases for the first row, 8 cases in the second and so forth – avoid such potential confusion.

..if ever in Tab 2 you mean Nb as case ID. Am finally not sure what you mean by “occurrence of barriers among the 25 case-studies”, row one: it occurs in case 21 or it occurred 21 times? If you mean 21 times then better use “n”. If you mean case 21 then better use something like ID.

L372 what are “contrasted” positive impacts? May be you meant complementary or cumulative? Or simply positive impacts is enough?

L382 the use of “external” inputs? Cycling of internal goods is quite useful I think, you probably mean less dependency on inputs to be bought.

Reviewer #3: I have carefully read the responses to the comments of the 3 reviewers and I congratulate the authors for the work they have done to take into account these comments. The paper is now much more convincing following the enrichment of the methods and discussion sections and it also seems to me to be better structured and easier to read. The objective of the paper is also more in adequacy with all the materials presented.

The strengthening of section 2.2 with the additions of Figure 3 and Appendix S1 is welcome. The method of data collection and analysis is now clearly detailed with also details of the limitations encountered. These explanations and limitations came far too late in the previous version.

Remarks on "food system innovation settings" based on 3 criteria with 3 modalities have been mainly addressed. We now know better why and how the authors chose these criteria and how these criteria can be combined with each other and lead to specific configurations. And it was important to specify in the article that the categories are based on an approach that combines existing literature and prior qualitative analysis of the interviews.

The beginning of the results section is now much clearer thanks to the reorganizations that have been made.

The addition of the comparative dimension in section 4.3 is welcomed with the emphasis on the links between the configuration of food systems and the barriers encountered, which effectively opens up perspectives on appropriate policies to remove these barriers.

The hybridization of food systems is now treated in a more objective and modest way with regard to the materials used (the diversity of barriers encountered depending on the nature of the food system considered). The addition of a section 4 in the discussion makes it possible to give consistency to this hybridization in connection with the work on the "intermediate value chain". It is also interesting to note that the strong links between local chains and organic and between conventional farming and commodity value chains contribute to the question of system hybridization due to the development of AB and cyclical crises in commodity markets.

Thanks you for the redesign of the conclusion

The remarks on Table 1, Figure 2 and Figure 3 were also addressed.

The addition of the paragraph on the role that researchers can play in the dynamics of innovation seemed to me to be very relevant (606-620)

typo

217 : I suggest removing the 5 in front of « different innovation teams »

696-698 : there are two times the same verb « are growing » in the sentence

707-708 : to be rephrased « hybrid configurations result in solving OF creating more barriers »

7. PLOS authors have the option to publish the peer review history of their article (what does this mean?). If published, this will include your full peer review and any attached files.

Reviewer #1: No

Reviewer #3: No

---

## [Author Response · Author response to Decision Letter 1]

14 Feb 2020

Dear editors and reviewers,

We thank you for your positive feedbacks on the changes we made based on your initial comments. Please find below, the new changes we made to address your “second round” of comments.

Please consider all relevant language issues raised by the two reviewers. It is recommended to consult a professional native speaker for general language editing to further improve the clarity of your manuscript.

==>The whole manuscript has been proofread and edited by a native professional English translator. 

Reviewer #1: The authors have well responded to the comments. Still, a thorough language check is needed. The text is comprehensible but there are still so many incidences where the text is not clear enough, the used terms probably not always the correct ones or where grammar issues cause some confusion (e.g. prepositions, the use of plural/singular in composed terms, commas, using “the”, and the s in the present tense). In the following some detailed comments – but I stopped with marking language issues after some lines.

==>The whole manuscript has been proofread and edited by a native professional English translator.

L26-28 instead of always writing “a” type of, use “the” type of

==>We have replaced “the type” by “a type” along the manuscript.

L32 where crop diversification increases (with s)

==>We have added a “s”. 

L43: The transition … requires (with s)

==>We have added a “s”.

L73: of farmers taking the related risks (not whatsoever risks)

==>We have added “the related risks”.

L82: approaches (with s)

==>We have added a “s”.

L89-92: complicated sentence, check with a native

==>The sentence was edited by a native.

L101-103: be more specific, check the correct wording. Why is “reduced inputs” a sustainability benefit? Did you mean something like relying less on external inputs? “yields gaps” – closing yield gaps? I think the s from risks is not needed here. Normally we use the term ecosystem services – did you mean this by ecosystemic services, or is that something else?

==>Wording was checked by a native. We added “reduced” yield gaps to be more specific and “ecosystem services” instead of “ecosystemic services”. “Reduced inputs” was replaced by “reduced dependency on external inputs” to be clearer. 

L103: have proven

==>We have replaced it by “have shown”.

L104: of “the” food systems

….am more or less stopping here with my specific language, grammar etc notes…

==>The native proof-reader judged that “barriers at all levels of food systems” was ok. The whole text was edited by this native proof-reader. 

L119: lens? Is that lentils?

==>We meant lentils and changed the wording accordingly. 

Tab 1 (and Tab2): in the first row of Tab 1, better use Nb as used in Tab 2, good for consistency, instead of number. Even though the legend explains it, number may cause people think that you talk about the number of cases, such as 3 cases for the first row, 8 cases in the second and so forth – avoid such potential confusion.

..if ever in Tab 2 you mean Nb as case ID. Am finally not sure what you mean by “occurrence of barriers among the 25 case-studies”, row one: it occurs in case 21 or it occurred 21 times? If you mean 21 times then better use “n”. If you mean case 21 then better use something like ID.

==>In Tab 1, the number is the case ID, so we changed “Case number” by “Case ID”. In the legend of Tab 1, we now specify “The cases are called by their ID in the DiverIMPACTS project, to make it easier to find more information on the project website: https://www.diverimpacts.net/case-studies.html””

In Tab 2, the number relates to the number of times the barrier occured, so as suggested we use now « n” and in the legend indicate: “number of occurrences of the related barrier among the 25 case-studies”. 

This clarification was indeed very useful!

L372 what are “contrasted” positive impacts? May be you meant complementary or cumulative? Or simply positive impacts is enough?

==>Positive impacts is enough and we deleted “contrasted”. 

L382 the use of “external” inputs? Cycling of internal goods is quite useful I think, you probably mean less dependency on inputs to be bought.

==>We replaced it by “could reduce the dependency on external inputs”. 

Reviewer #3: 

217 : I suggest removing the 5 in front of « different innovation teams »

==>Here we wanted to explain that we organised 5 different workshops and that each workshop involved 5 innovation teams. So we modified the sentence to be clearer: “In a first round, 5 workshops were organised, each involving 5 different innovation teams out of 25. “

696-698 : there are two times the same verb « are growing » in the sentence

==>We corrected that and the native editor reformulated the sentence as such: “Industrialised countries are witnessing the growth of intermediate food value chains”

707-708 : to be rephrased « hybrid configurations result in solving OF creating more barriers »

==>We rephrased it as such: ““hybrid configurations remove or create more barriers”.

---

## [Editor Report · Decision Letter 2]

19 Feb 2020

Innovating within or outside dominant food systems? Different challenges for contrasting crop diversification strategies in Europe

PONE-D-19-17348R2

Dear Dr. Morel,

We are pleased to inform you that your manuscript has been judged scientifically suitable for publication and will be formally accepted for publication once it complies with all outstanding technical requirements.

With kind regards,

Til Feike, PhD

Academic Editor

PLOS ONE
---

## [Editor Report · Acceptance letter]

26 Feb 2020

PONE-D-19-17348R2 

Innovating within or outside dominant food systems? Different challenges for contrasting crop diversification strategies in Europe 

Dear Dr. Morel:

I am pleased to inform you that your manuscript has been deemed suitable for publication in PLOS ONE. Congratulations! Your manuscript is now with our production department. 

With kind regards,

on behalf of

Dr. Til Feike 

Academic Editor

PLOS ONE